# Fe$_3$O$_4$-doped mesoporous carbon cathode with a plumber's nightmare structure for high-performance Li-S batteries

Han Zhang[1,2], Mengtian Zhang[3], Ruiyi Liu[1,2], Tengfeng He[4], Luoxing Xiang[1,2], Xinru Wu[3], Zhihong Piao[3], Yeyang Jia[3], Chongyin Zhang[4], Hong Li[3] ✉, Fugui Xu [1,2] ✉, Guangmin Zhou [3] ✉ & Yiyong Mai [1,2] ✉

Shuttling of lithium polysulfides and slow redox kinetics seriously limit the rate and cycling performance of lithium-sulfur batteries. In this study, Fe$_3$O$_4$-doped carbon cubosomes with a plumber's nightmare structure (SP-Fe$_3$O$_4$-C) are prepared as sulfur hosts to construct cathodes with high rate capability and long cycling life for Li-S batteries. Their three-dimensional continuous mesochannels and carbon frameworks, along with the uniformly distributed Fe$_3$O$_4$ particles, enable smooth mass/electron transport, strong polysulfides capture capability, and fast catalytic conversion of the sulfur species. Impressively, the SP-Fe$_3$O$_4$-C cathode exhibits top-level comprehensive performance, with high specific capacity (1303.4 mAh g$^{-1}$ at 0.2 C), high rate capability (691.8 mAh gFe$_3$O$_4^1$ at 5 C), and long cycling life (over 1200 cycles). This study demonstrates a unique structure for high-performance Li-S batteries and opens a distinctive avenue for developing multifunctional electrode materials for next-generation energy storage devices.

The multielectron conversions of sulfur atoms endow lithium-sulfur (Li-S) batteries with superior theoretical specific capacity (1675 mAh g$^{-1}$)[1–3]. Moreover, sulfur is abundant, inexpensive, and environmentally friendly[4,5]. These distinguishing features make them promising for use in Li-S batteries for consumer electronic products[6,7]. However, several unsolved problems hinder the realization of their improved performance and practical application. First, sulfur and most of the lithium polysulfides (LiPSs) are insulators, which leads to slow redox kinetics and limited active sulfur utilization[6,8]. Second, shuttling of the lithium polysulfides (consisting of Li$_2$S$_8$, Li$_2$S$_6$, and Li$_2$S$_4$) and the slow kinetics for LiPSs conversion result in rapid capacity fading and a loss of coulombic efficiency[4,9–11]. Third, the large volume difference between S and Li$_2$S causes shedding of the active materials from the current collector in the charge/discharge process[12–14].

In this context, a number of strategies have been developed to address these obstacles[15–17]. Notably, the construction of multifunctional sulfur hosts is an efficient approach to improve the performance of Li-S batteries by taking advantage of their flexible structural/functional designability[18–20]. Among the multiple developed sulfur hosts, porous carbon particles (PCPs) are attractive candidates owing to their high conductivity, high porosity/specific surface areas (SSAs), and high structural stability[3,21]. However, the available PCP-based Li-S batteries still do not meet the ever-increasing demand. The main reasons for this difference include: (1) most PCPs only have micropores or obstructed pores, which makes it difficult for sulfur and electrolytes to penetrate deeply into their interior[22–24]; (2) hindered mass transport in the interiors of thick electrodes severely limits the contact between the LiPSs and the internal active matrix of PCPs, leading to poor sulfur utilization[25,26]; (3) the absence of polar active

[1]School of Chemistry and Chemical Engineering, Shanghai Jiao Tong University, 800 Dongchuan Road, Shanghai 200240, China. [2]Frontiers Science Center for Transformative Molecules, Shanghai Jiao Tong University, 800 Dongchuan Road, Shanghai 200240, China. [3]Tsinghua-Berkeley Shenzhen Institute & Tsinghua Shenzhen International Graduate School, Tsinghua University, Shenzhen 518055, China. [4]Shanghai Aerospace Equipments Manufacturer Co., Ltd., 100 Huaning Road, Shanghai 200245, China. ✉e-mail: melihong@alumni.sjtu.edu.cn; xufg1227@sjtu.edu.cn; guangminzhou@sz.tsinghua.edu.cn; mai@sjtu.edu.cn

sites in PCPs leads to weak adsorption and slow catalytic conversion of the LiPSs. Therefore, the introduction of 3D continuous open channels and polar species (e.g., metal oxides) into PCPs is a desirable strategy for solving these problems. Recently, bicontinuous mesoporous structures have attracted much attention for use in energy storage applications[27–29], because these structures provide 3D continuous frameworks and open mesochannels (Fig. 1a for illustration), which enable smooth mass transport into the deep internal areas of electrode materials and thus remarkably increase the accessibility of the inner active sites[30,31]. However, the construction of bicontinuous architectures is highly challenging due to their complex curved channels; for example, double or single primitive categories are called plumber's nightmare structures[32–34]. The currently available techniques, such as photolithography, 3D printing and molecular self-assembly, are still difficult to produce these structures, especially at the mesoscale (2–50 nm)[35–39]. Therefore, bicontinuous mesoporous materials has rarely been used in energy storage applications[40,41]; their contributions to Li-S battery performance have remained unexplored, which inspires the interest of study.

Here, we report the preparation of $Fe_3O_4$-dopped carbon cubosomes (named SP-$Fe_3O_4$-C) with a single-network plumber's nightmare structure (i.e., single primitive (SP) bicontinuous topology). The synthetic strategy employed polymer cubosomes (PCs; cubosomes generally refer to colloidal particles with bicontinuous cubic liquid crystal phases) as the removable template and a biomass-derived Fe metal-phenolic network (Fe-MPN) as the $Fe_3O_4$ and carbon precursor (Fig. 1). The PCs were obtained by self-assembly of a block copolymer (BCP) in solution through precise adjustment of the self-assembly parameters. The Fe-MPN was utilized for three reasons: (1) its precursors, plant polyphenols, are natural and nontoxic biomasses[42–44]; (2) the Fe ions are uniformly coordinated in the network and can be directly (one-step) converted into $Fe_3O_4$ nanoparticles evenly doped in a carbon matrix after the pyrolysis of Fe-MPN, avoiding the complicated steps of traditional postloading methods for $Fe_3O_4$ particles[45–47]; and (3) the uniformly distributed $Fe_3O_4$ nanoparticles possess certain conductivity ($5 \times 10^4$ S m⁻¹) which is sufficient for charge transfer during battery operation and serve as active sites to capture the LiPSs and accelerate their catalytic conversions[20,48,49]. Based on these merits, the

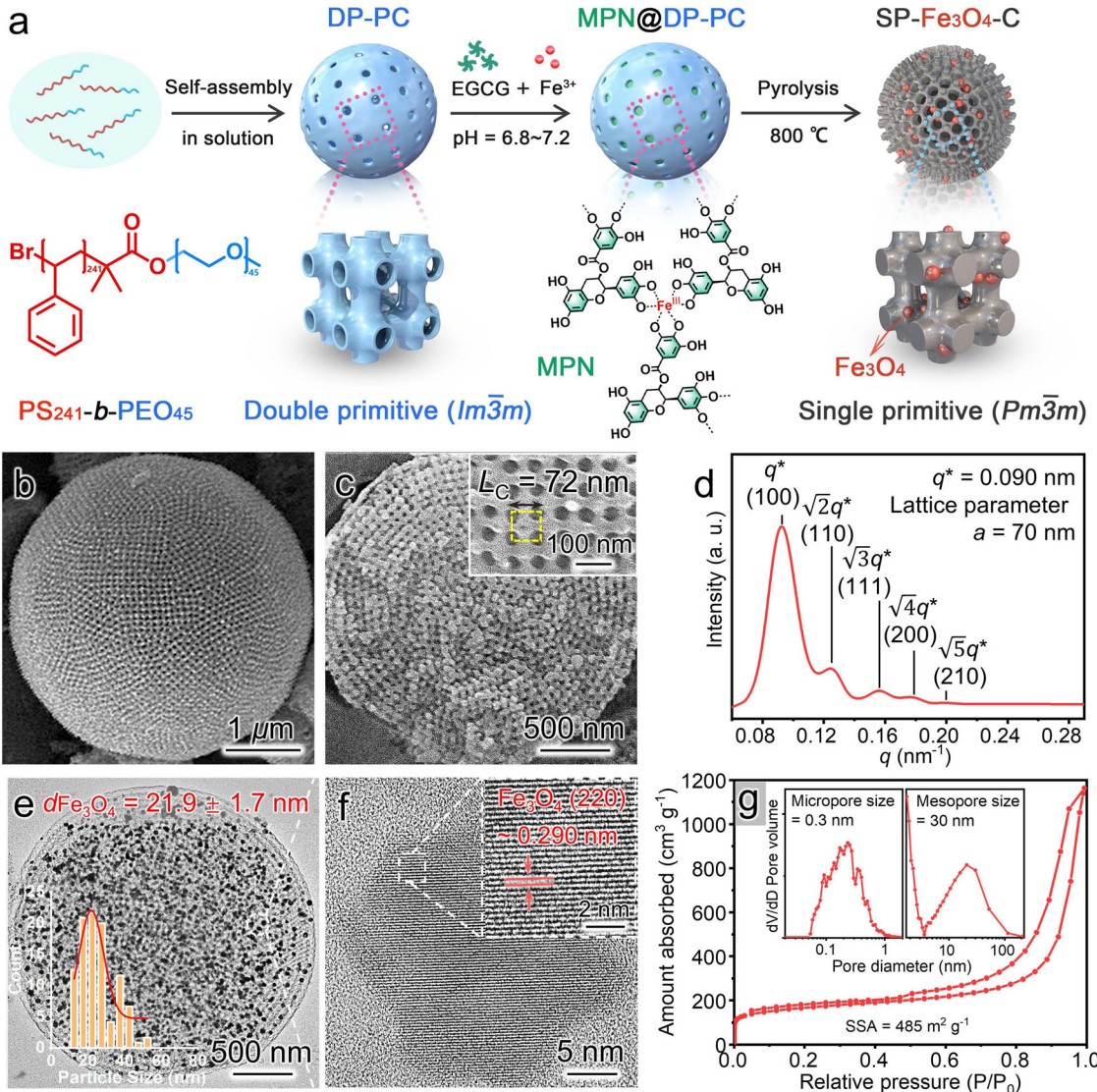

**Fig. 1 | Structural characterizations of SP-$Fe_3O_4$-C. a** Schematic preparation of SP-$Fe_3O_4$-C with a plumber's nightmare (bicontinuous) structure. **b** SEM image of an SP-$Fe_3O_4$-C cubosome along the (100) direction. **c** SEM image of a fractured SP-$Fe_3O_4$-C cubosome. The inset shows a high-magnification SEM image of a local area. **d** SAXS pattern of the SP-$Fe_3O_4$-C powder. **e** TEM image of an SP-$Fe_3O_4$-C cubosome and the size distribution curve for the $Fe_3O_4$ particles (inset). **f** HRTEM image of a $Fe_3O_4$ nanoparticle in SP-$Fe_3O_4$-C. The inset presents a high-magnification SEM image of a local area. **g** Nitrogen adsorption-desorption isotherm and pore size distribution of SP-$Fe_3O_4$-C (inset).

resultant SP-Fe$_3$O$_4$-C had a high Fe$_3$O$_4$ loading content of 19.5 wt% and a large SSA of 485 m$^2$ g$^{-1}$. The Li-S batteries with the SP-Fe$_3$O$_4$-C/S cathodes (~ 75 wt% sulfur loading) exhibited a high initial discharge capacity of 1303.4 mAh g$^{-1}$ at 0.2 C, impressive rate performance (691.8 mAh g$^{-1}$ at 5C), and high cycling stability (capacity retention rate over 68.1% and a 0.027% decay for each cycle over 1200 cycles at 1 C) and a high areal capacity of 6.5 mAh cm$^{-2}$ at a sulfur loading of 8.2 mg cm$^{-2}$. This comprehensive performance ranks among the top-level for reported carbon-based cathodes for Li-S batteries. The contributions of the bicontinuous structure and the Fe$_3$O$_4$ particles, along with the related mechanisms, were unveiled, which will help in the design and preparation of multifunctional cathode materials for Li-S batteries.

## Results and discussion

### Structural characterizations of SP-Fe$_3$O$_4$-C and SP-Fe$_3$O$_4$-C/S

The synthetic route toward SP-Fe$_3$O$_4$-C is illustrated in Fig. 1a. First, polymer cubosomes with an ordered double primitive topology (DP-PCs) were prepared via self-assembly of polystyrene-*block*-poly(-ethylene oxide) (PS$_{241}$-*b*-PEO$_{45}$) in solution following previously reported procedures (see details in the Supplementary Information, SI)[50]. The resultant PCs had a unit cell parameter (*a*) of *a* = 92 nm (Supplementary Figs. 1–3). The DP-PCs were subsequently used as templates to prepare the carbon cubosomes by employing (-)-epi-gallocatechin gallate (EGCG) and Fe$^{3+}$ as functional precursors. Typically, in a mixed aqueous solution with a pH of 6.8–7.2, Fe$^{3+}$ ions and EGCG molecules diffused into the open channels of the DP-PCs and formed MPN@DP-PC composite particles after coordination. The resultant MPN@DP-PCs were collected via centrifugation (the details are given in the SI). As shown in the scanning electron microscopy (SEM) and transmission electron microscopy (TEM) micrographs (Supplementary Fig. 4a, b), the mesochannels within the DP-PCs were fully filled with MPNs. The Fourier transform infrared (FTIR) spectrum demonstrated that the stretching vibration peak for the carbonyl group (C=O) in EGCG shifted from 1691 cm$^{-1}$ to 1640 cm$^{-1}$ in MPN@DP-PCs, indicating the presence of strong hydrogen bonds between the EGCG ligands and the PC templates (Supplementary Fig. 4c)[51]. Moreover, broadening of the HO-C stretching vibrational peaks of EGCG (3349 cm$^{-1}$ and 3471 cm$^{-1}$) indicated that the phenolic hydroxyl groups formed coordination bonds with the metal ions (Supplementary Fig. 4c)[51].

The SP-Fe$_3$O$_4$-C particles were obtained by pyrolysis of the resultant MPN@DP-PCs at 350 °C and 800 °C for 2 h, respectively; in this process, the DP-PC templates were also removed. The molar ratio of EGCG to Fe$^{3+}$ ($n_{EGCG}$/$n_{Fe3+}$) impacted the final morphology of the obtained carbon particles (see Supplementary Fig. 5 and details in the SI). Thereafter, SP-Fe$_3$O$_4$-C was synthesized with the optimized molar ratio of $n_{EGCG}$/$n_{Fe3+}$ = 1.3. TEM images revealed that the SP-Fe$_3$O$_4$-C particles had porous spherical structures with a mean size of 2.3 ± 0.4 $\mu$m (Fig. 1b and Supplementary Fig. 6). The cross-sectional SEM image of SP-Fe$_3$O$_4$-C confirmed the ordered mesoporous structure with a square lattice (Fig. 1c), while the high-magnification local SEM image clearly showed an SP cubic structure (inset of Fig. 1c). The mean side length of the SP-Fe$_3$O$_4$-C cubic unit ($L_C$ = 72 nm) was shorter than that of the PCs ($L_t$ = 90 nm) owing to shrinkage of the SP skeleton during carbonization. The average frame and pore sizes of SP-Fe$_3$O$_4$-C were 33.2 ± 3.1 nm and 30.5 ± 3.8 nm, respectively (Supplementary Fig. 7). Moreover, the topology of SP-Fe$_3$O$_4$-C was also studied by small-angle X-ray scattering (SAXS). As shown in Fig. 1d, the appearance of five characteristic peaks confirmed the SP structure of SP-Fe$_3$O$_4$-C (with a $Pm\bar{3}m$ symmetry)[27]. The unit cell parameter of SP-Fe$_3$O$_4$-C was *a* = 70 nm, which was close to the length of the cubic units ($L_C$ = 72 nm) measured from the SEM image (Fig. 1c). The TEM images revealed that SP-Fe$_3$O$_4$-C had an internal interconnected porous structure with clear frameworks and mesopores (Fig. 1e). Apparently,

numerous nanoparticles were homogeneously embedded in the carbon skeleton of SP-Fe$_3$O$_4$-C, and the average particle diameter was approximately 22 nm (inset of Fig. 1e). The high-resolution TEM (HRTEM) micrograph showed clear lattice fringes for the (220) plane in Fe$_3$O$_4$ (*d*-spacing of 0.290 nm; Fig. 1f)[52,53]. The formation of Fe$_3$O$_4$ in the SP-Fe$_3$O$_4$-C was also confirmed by the X-ray diffraction (XRD) pattern, in which all of the diffraction peaks were indexed to the Fe$_3$O$_4$ phase (JCPDS No. 88-0866, Supplementary Fig. 8a)[54]. Nitrogen adsorption-desorption analysis of SP-Fe$_3$O$_4$-C yielded a type IV isotherm with an H$_1$-type hysteresis loop, validating the existence of mesopores (Fig. 1g)[55,56], while the uptake at low pressures in the isotherm suggested the coexistence of micropores[57]. The average mesopore size was ~ 30 nm, which was consistent with the SEM results. Moreover, the micropores within the SP-Fe$_3$O$_4$-C sample had an average diameter of 0.3 nm. Additionally, the broad pore size distribution of 8–100 nm was probably caused by random stacking of the SP-Fe$_3$O$_4$-C particles[58]. The total and micropore volumes of the SP-Fe$_3$O$_4$-C composite were 1.08 cm$^3$ g$^{-1}$ and 0.11 cm$^3$ g$^{-1}$, respectively. The SSA of SP-Fe$_3$O$_4$-C was 485 m$^2$ g$^{-1}$. The porosity and surface area information of the SP-Fe$_3$O$_4$-C composite are presented in Supplementary Table. 1.

The Raman spectrum of SP-Fe$_3$O$_4$-C displayed two peaks attributed to the G (1589 cm$^{-1}$) and D (1345 cm$^{-1}$) bands (Supplementary Fig. 8b)[59]. A high G/D band intensity ratio ($I_G$/$I_D$~ 1.2) indicated a high degree of graphitization of the carbon framework, which favors electrical conductivity[60]. The Fe$_3$O$_4$ content of SP-Fe$_3$O$_4$-C was 19.5 wt% (Supplementary Fig. 8c) according to thermogravimetric analysis (TGA); this was confirmed with inductively coupled plasma–mass spectrometry (ICP–MS), which revealed an iron content of 13.8 wt% (i.e., a Fe$_3$O$_4$ content of 19.1 wt%). Moreover, X-ray photoelectron spectroscopy (XPS) demonstrated the presence of Fe, C, and O (Supplementary Fig. 9). The high-resolution Fe 2*p* spectrum was resolved into four main peaks at 726.8, 724.3, 713.8 and 711.3 eV, confirming the generation of Fe$_3$O$_4$ particles[49].

The 3D continuous pore channels, highly graphitized carbon framework, and homogeneously distributed Fe$_3$O$_4$ nanoparticles make SP-Fe$_3$O$_4$-C a promising candidate sulfur host material. SP-Fe$_3$O$_4$-C-based cathodes were prepared by the sulfur melt-diffusion method, which yielded SP-Fe$_3$O$_4$-C/S composites (details in the Methods section). The channels of SP-Fe$_3$O$_4$-C were almost completely filled with sulfur according to the TEM images of SP-Fe$_3$O$_4$-C (Supplementary Fig. 10a, c) and SP-Fe$_3$O$_4$-C/S (Supplementary Fig. 10b, d), which demonstrated the successful sulfur loading of SP-Fe$_3$O$_4$-C. The homogeneous distribution of sulfur was shown with energy-dispersive X-ray spectroscopy (EDS) elemental maps (Supplementary Fig. 10e). According to the TGA measurements, the sulfur content in the SP-Fe$_3$O$_4$-C/S cathode was approximately 75 wt% (Supplementary Fig. 10f). In addition, the XRD pattern of SP-Fe$_3$O$_4$-C/S confirmed the successful loading of long sulfur chains with an orthorhombic structure (S$_8$, JCPDS no. 77-0145, Supplementary Fig. 10g)[61]. Moreover, the SSA and pore volume of SP-Fe$_3$O$_4$-C/S sharply decreased to 49 m$^2$ g$^{-1}$ and 0.31 cm$^3$ g$^{-1}$, respectively, indicating the encapsulation of sulfur (Supplementary Fig. 10h, i; Supplementary Table. 1). It should be noted that the adsorption/desorption isotherm and the pore size distribution also demonstrated the presence of a little spare pore space inside SP-Fe$_3$O$_4$-C/S, which could provide additional tunnels for mass diffusion (Supplementary Fig. 10h, i).

As a control sample, Fe$_3$O$_4$ nanoparticle doped carbon particles with discontinuous mesopores were also prepared and labelled B-Fe$_3$O$_4$-C (Supplementary Fig. 11a). As revealed by SEM, the B-Fe$_3$O$_4$-C particles were irregular (Supplementary Fig. 11b). A high-magnification TEM image demonstrated the presence of discontinuous pores and embedded Fe$_3$O$_4$ nanoparticles in B-Fe$_3$O$_4$-C (Supplementary Fig. 11c). The XRD (Supplementary Fig. 8a), Raman (Supplementary Fig. 8b) and XPS (Supplementary Fig. 12) spectra of B-Fe$_3$O$_4$-C were similar to those of SP-Fe$_3$O$_4$-C. The ferric oxide content of B-Fe$_3$O$_4$-C was 21.0 wt%

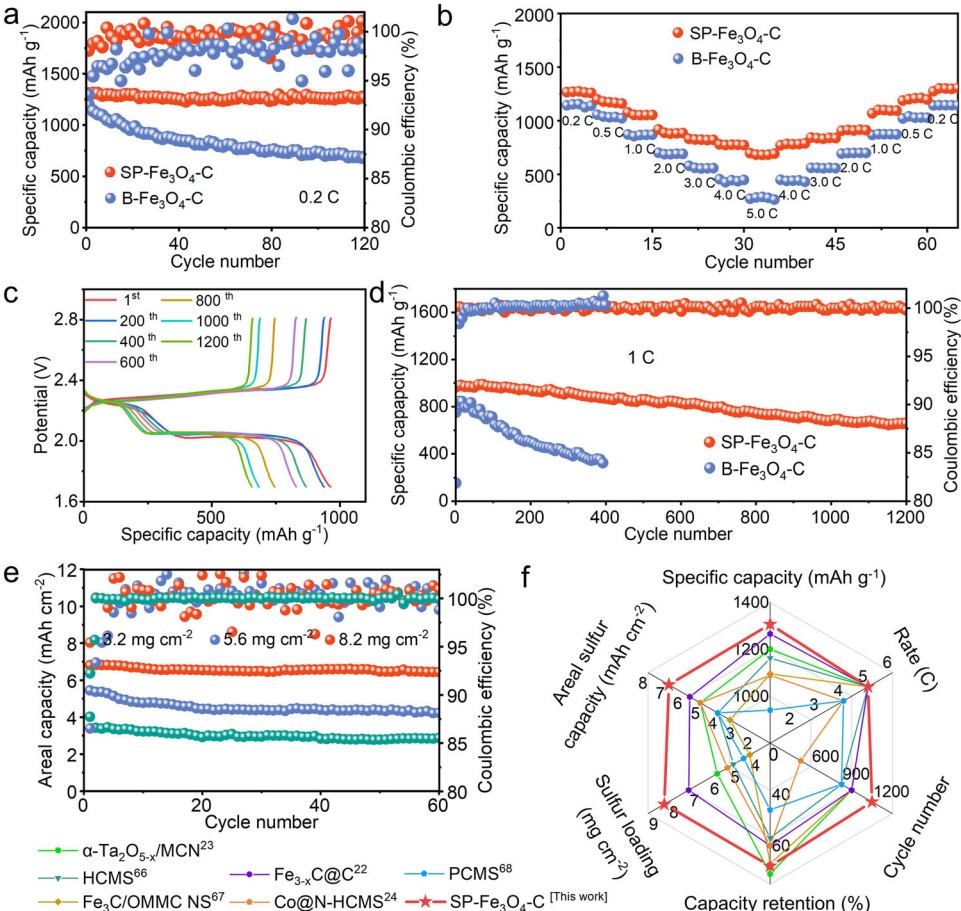

**Fig. 2 | Electrochemical comparison of the SP-Fe₃O₄-C/S-based and B-Fe₃O₄-C/S-based Li-S batteries. a** Cycling performance at 0.2 C. **b** Rate performance. **c** GCD curves of the SP-Fe₃O₄-C/S-based Li-S batteries from the 1st to 1200th cycles at 1 C. **d** Long cycling performance at 1 C. **e** Cycling performance of the SP-Fe₃O₄-C/S-based Li-S batteries with sulfur loadings of 3.2, 5.6 and 8.2 mg cm⁻². **f** Performance radar chart of the SP-Fe₃O₄-C and PCP cathode-based Li-S batteries with reported state-of-the-art examples.

based on TGA (Supplementary Fig. 8c), which was close to that of SP-Fe₃O₄-C. The SSA of B-Fe₃O₄-C was 418 m² g⁻¹ (Supplementary Fig. 13a), which was slightly smaller than that of SP-Fe₃O₄-C, probably due to discontinuous pores leading to impalpable internal surfaces. The micropores within the B-Fe₃O₄-C sample had an average diameter of 0.8 nm. The total and micropore volumes of the B-Fe₃O₄-C composite were 0.64 cm³ g⁻¹ and 0.10 cm³ g⁻¹, respectively. The control sample was also loaded with sulfur via the melt-diffusion method, producing B-Fe₃O₄-C/S. The sulfur content of B-Fe₃O₄-C/S was 74 wt% (Supplementary Fig. 13b), close to that of SP-Fe₃O₄-C/S (75 wt%). The SSA of B-Fe₃O₄-C/S was much lower (1 m² g⁻¹), confirming the successful loading of sulfur (Supplementary Fig. 13c, d), and other specific parameters of B-Fe₃O₄-C/S were listed in Supplementary Table. 1.

**Performance of Li-S batteries with the SP-Fe₃O₄-C/S cathode**

The electrochemical properties of the SP-Fe₃O₄-C/S cathode were tested systematically with assembled CR2032 Li-S cells. To verify the structural advantages of SP-Fe₃O₄-C/S for Li-S batteries, the electrochemical properties of the SP-Fe₃O₄-C/S and B-Fe₃O₄-C/S cathodes were compared (Fig. 2). At a low current of 0.2 C, the SP-Fe₃O₄-C/S cathode exhibited a higher initial discharge capacity of 1303.4 m Ah g⁻¹ than the B-Fe₃O₄-C/S cathode (1196.4 mAh g⁻¹). Moreover, the SP-Fe₃O₄-C/S cathode exhibited stable cycling performance with a lower capacity decay of only 0.024% per cycle after 120 cycles (Fig. 2a). In contrast, the B-Fe₃O₄-C/S cathode underwent a very fast capacity decay, with 57% of the capacity retained after 120 cycles, which

corresponded to a much greater capacity decay of 0.35% per cycle (Fig. 2a). The corresponding galvanostatic charge/discharge (GCD) curves from the 1st to 120th cycles (Supplementary Fig. 14) showed that, compared with those of the B-Fe₃O₄-C/S cathode, the SP-Fe₃O₄-C/S cathode exhibited two well-defined longer discharge plateaus even after 120 cycles[62]. This result demonstrated the enhanced sulfur redox kinetics, especially for the two redox processes by which solid $S_8$ was converted to highly soluble $Li_2S_8$ and soluble $Li_2S_4$ was converted to insoluble $Li_2S_2/Li_2S$ in the whole sulfur redox process[63]. During cycling of the SP-Fe₃O₄-C/S cathode, the SP-Fe₃O₄-C/S particles were uniformly dispersed in the cathode matrix (Supplementary Fig. 15a). After cycling, there was almost no sulfur aggregation on the surface of the SP-Fe₃O₄-C/S cathode (Supplementary Fig. 15b). Moreover, the bicontinuous structure was retained after repeated charge/discharge cycles (Supplementary Fig. 15c, d), indicating the good structural stability of the SP-Fe₃O₄-C/S composite. In contrast, the significant morphological changes in the cycled B-Fe₃O₄-C/S cathode revealed its poor stability (Supplementary Fig. 16).

The rate performance was evaluated and is presented in Fig. 2b. The SP-Fe₃O₄-C/S cathode demonstrated superior rate performance, with capacities of 1303.4, 1197.0, 1093.9, 911.0, 837.4 and 788.8 mAh g⁻¹ at 0.2, 0.5, 1.0, 2.0, 3.0 and 4.0 C, respectively; even at a high rate of 5 C, it still delivered a high capacity of 691.8 mAh g⁻¹ (Fig. 2b). The rate capacity of the SP-Fe₃O₄-C/S solution was much greater than that of the B-Fe₃O₄-C/S solution at all testing rates. Impressively, the capacity of SP-Fe₃O₄-C/S was restored to 1273.8 mAh g⁻¹ after the current was

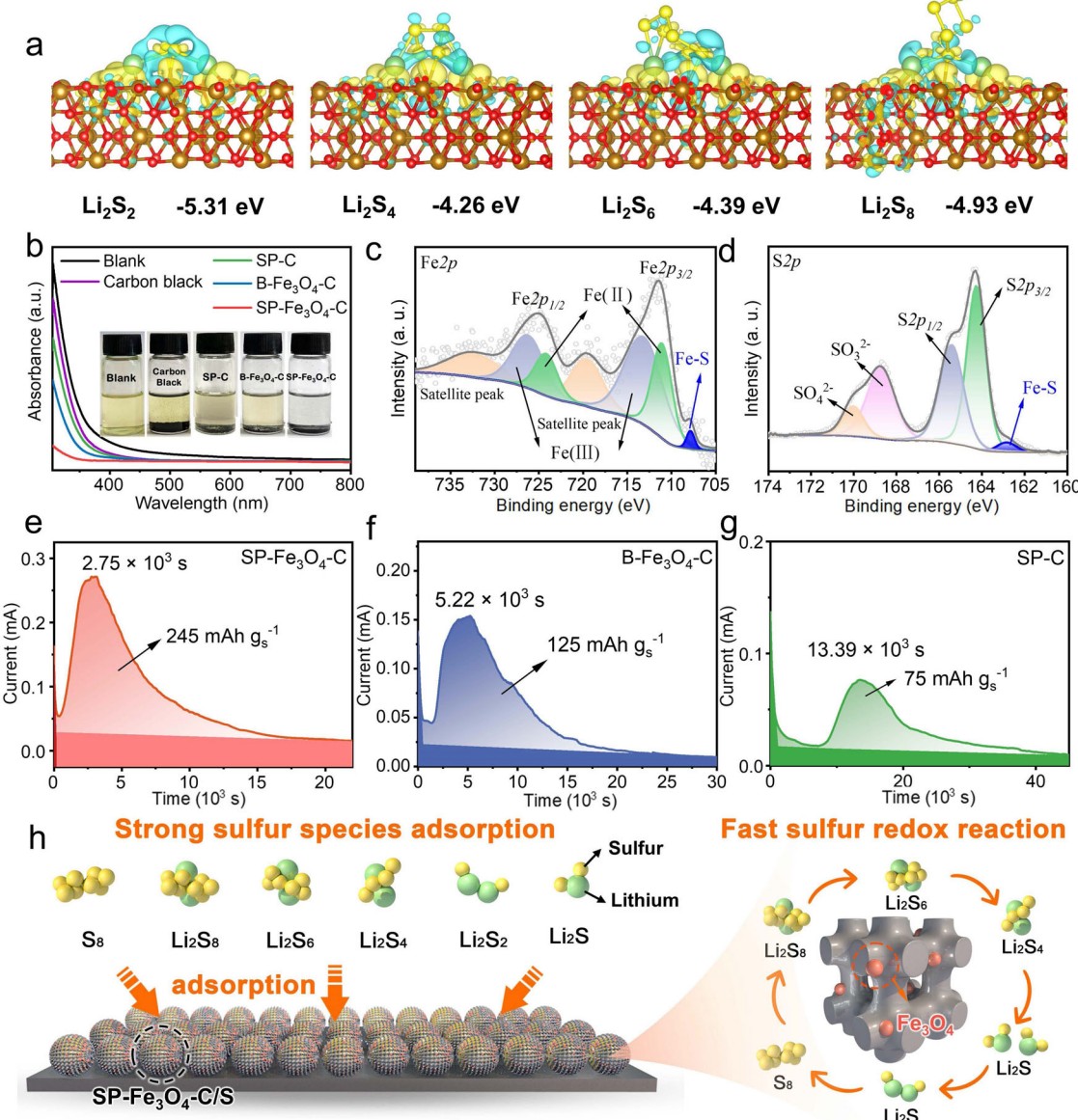

**Fig. 3 | Characterization of the effect of Fe₃O₄. a** DFT-calculated energies for adsorption of different LiPSs on $Fe_3O_4$ (220). Lithium, sulfur, iron, and oxygen atoms are marked with green, yellow, brown and red, respectively. The yellow and blue regions represent increases and decreases in charge density, respectively. The adsorption energies are given under each structure. **b** UV–vis spectra of $Li_2S_6$ adsorbed by different samples. The insets show photographs of $Li_2S_6$ adsorption by

different samples: blank, carbon black, SP-C, B-Fe₃O₄-C and SP-Fe₃O₄-C. **c** High-resolution Fe $2p$ XPS spectrum of the cycled SP-Fe₃O₄-C/S composite. **d** High-resolution S$2p$ XPS spectrum of the cycled SP-Fe₃O₄-C/S composite. **e**–**g** Li₂S pre-cipitation experiments of SP-Fe₃O₄-C, B-Fe₃O₄-C and SP-C. **h** Schematic illustration of the adsorption and catalytic conversion of sulfur species.

returned to 0.2 C. The high rate performance of the SP-Fe₃O₄-C/S cathode indicated fast electron transport along the carbon framework and enhanced ion transport in the 3D mesopores during battery operation. The battery with the SP-Fe₃O₄-C/S cathode also exhibited high cycling capacity at 1 C (Fig. 2c, d). The SP-Fe₃O₄-C/S cathode had an initial capacity of 965.8 mAh g⁻¹ with an average coulombic effi-ciency of 99.94% over 1200 cycles. Notably, the SP-Fe₃O₄-C/S-based batteries exhibited a low capacity decay of 0.027% per cycle after 1200 cycles, outperforming the performance of the B-Fe₃O₄-C/S-based batteries. Because a high sulfur loading is required for practical application[64,65], a thick cathode with a high areal loading of sulfur was tested as well. At 0.1 C, stable cycling was seen with sulfur loads of 3.2, 5.6, and 8.2 mg cm⁻² (Fig. 2e). After 60 cycles, the Li-S battery with the SP-Fe₃O₄-C/S cathode had a high areal capacity of 6.5 mAh cm⁻² with an 8.2 mg cm⁻² areal loading of sulfur, which was much greater than those

of commercial Li-ion batteries (~ 4 mAh cm⁻²). Among those of all PCP-based[22–24,66–68] (Supplementary Table. 2 and Fig. 2f) and even carbon-based sulfur hosts (Supplementary Table. 3 and Supplementary Fig. 17) reported in the past five years, the comprehensive performance of the SP-Fe₃O₄-C/S-based Li-S battery ranked near the top.

## Exploring the effect of Fe₃O₄ on the performance of the SP-Fe₃O₄-C/S-based Li-S battery

The high electrochemical performance of SP-Fe₃O₄-C/S stemmed from a synergistic effect of the unique 3D interconnected open channels and the carbon frameworks with embedded Fe₃O₄ particles. The key roles of Fe₃O₄ included chemical adsorption and catalytic conversion of the LiPSs[62]. As a control sample, PCPs were prepared with an SP structure but without Fe₃O₄ particles (SP-C) (Supplementary Fig. 18). Density functional theory (DFT) calculations were used to understand the

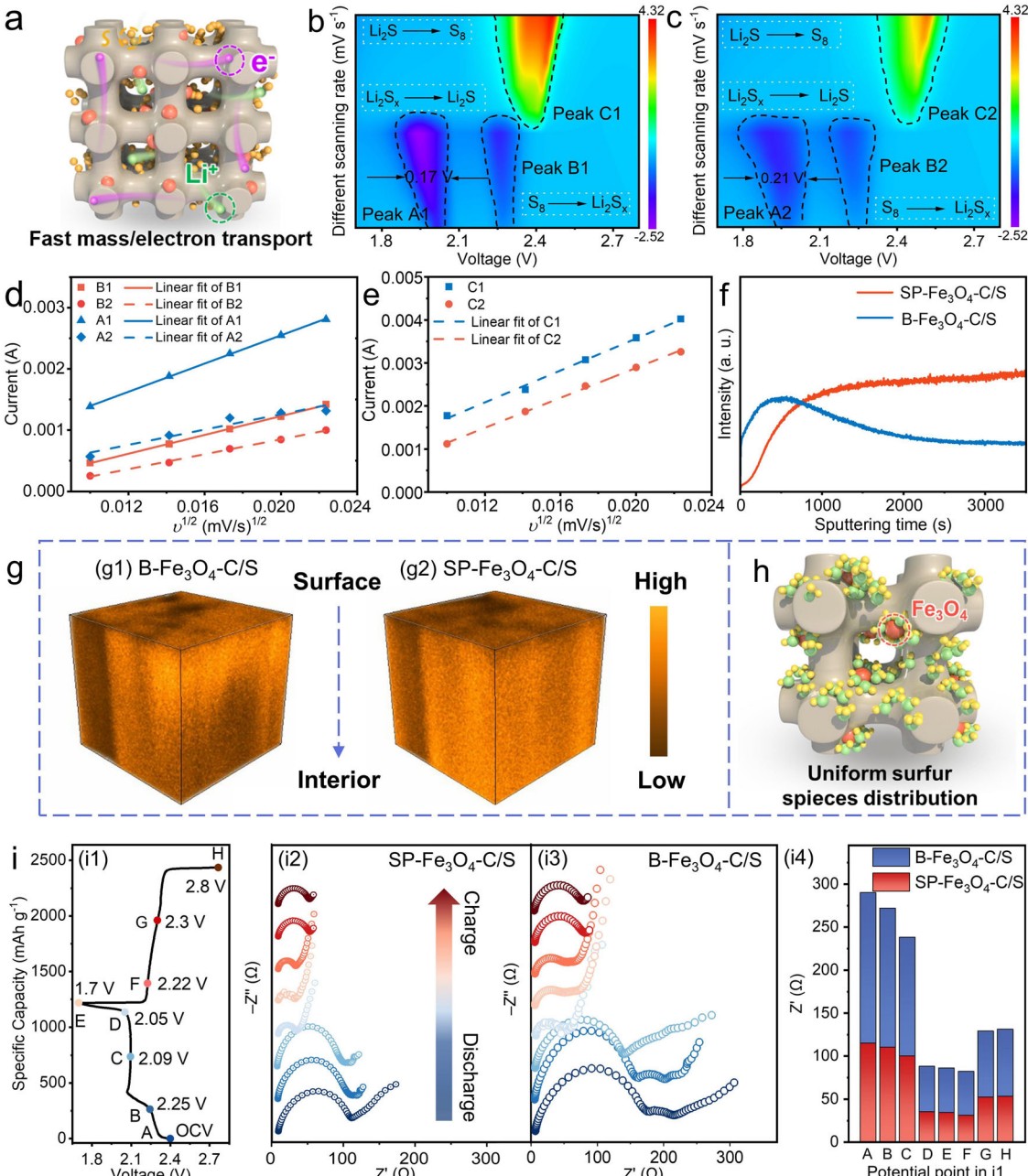

**Fig. 4 | Characterization of the effect of the bicontinuous structure. a** Schematic showing ion/electron transport in the continuous mesochannels of SP-Fe$_3$O$_4$-C/S; carbon frame, Fe$_3$O$_4$ nanoparticle, sulfur, electron, and lithium ion are marked with grain, red, yellow, pink, and green, respectively. **b** Contour maps for the CV curves of SP-Fe$_3$O$_4$-C/S (0.1 - 0.5 mV s$^{-1}$). **c** Contour maps for the CV curves of B-Fe$_3$O$_4$-C/S (0.1-0.5 mV s$^{-1}$). **d, e** Peak currents versus square root of scan rates of SP-Fe$_3$O$_4$-C/S (A1, B1, C1) and B-Fe$_3$O$_4$-C/S (A2, B2, C2). **f** Normalized depth profiles for the S$^-$ secondary ion fragments obtained by TOF-SIMS analyses of the SP-Fe$_3$O$_4$-C/S and B-Fe$_3$O$_4$-C/S cathodes after 50 cycles at 0.2 C. **g** 3D depth profiles of the B-Fe$_3$O$_4$-C/S (g1) and SP-Fe$_3$O$_4$-C/S (g2) cathodes. **h** Schematic for LiPSs capture and Li$_2$S deposition in SP-Fe$_3$O$_4$-C/S; carbon frame, Fe$_3$O$_4$, sulfur atom, and lithium atom are marked with grain, red, yellow, and green, respectively. **i** In situ EIS: voltage curve of the in situ EIS during the 1st cycle (i1); in situ EIS data for the SP-Fe$_3$O$_4$-C/S (i2) and B-Fe$_3$O$_4$-C/S (i3) cathodes during the 1st cycle; the corresponding $R_{ct}$ histograms of SP-Fe$_3$O$_4$-C/S and B-Fe$_3$O$_4$-C/S obtained from the in situ EIS at different states (i4).

chemical interaction between Fe$_3$O$_4$ and LiPSs. Figure 3a displays the optimized configurations of LiPSs on the polar Fe$_3$O$_4$ (220) plane, in which Li$^+$ interacted with the O atoms and S$_n^{2-}$ was bound to the Fe atoms on the Fe$_3$O$_4$ surface (220)[49]. The calculated binding energies for Li$_2$S$_2$, Li$_2$S$_4$, Li$_2$S$_6$, and Li$_2$S$_8$ on the Fe$_3$O$_4$ surface were −5.31, −4.26, −4.39 and −4.93 eV, respectively (Fig. 3a), which were much greater than those on graphene. These results were confirmed by the differential charge densities of Fe$_3$O$_4$ and LiPSs, in which the charge density of the Fe atom increased while that of the S atom decreased, and these

results supported a chemical adsorption interaction between Fe$_3$O$_4$ and the LiPSs (Fig. 3a). The strong adsorption of the LiPSs may greatly alleviate migration during charging/discharging. Visualized adsorption experiments were carried out to evaluate the capacity of LiPSs adsorption on SP-Fe$_3$O$_4$-C. Typically, 10 mg of equivalent carbon black, SP-C, B-Fe$_3$O$_4$-C, or SP-Fe$_3$O$_4$-C was added to a Li$_2$S$_6$ solution (4 mM, 5 mL), after which the mixture was stirred. After 24 h, the mixture with SP-Fe$_3$O$_4$-C became nearly transparent after incubation (the inset of Fig. 3b). In sharp contrast, the samples containing the carbon black, SP-

C and B-Fe$_3$O$_4$-C still exhibited dark yellow colors. Moreover, the UV–vis absorption spectra of the supernatants were determined (Fig. 3b), and adsorption of the polysulfides decreased in the order: blank samples > carbon black >SP-C > B-Fe$_3$O$_4$-C > SP-Fe$_3$O$_4$-C. Notably, the polysulfide signal from the SP-Fe$_3$O$_4$-C sample was almost undetectable, indicating that the polysulfide adsorption capacity was greater than that of the control sample.

To study the mechanism for LiPSs conversion catalyzed by SP-Fe$_3$O$_4$-C, the XPS spectra of the cycled SP-Fe$_3$O$_4$-C/S cathode were recorded, as shown in Fig. 3c, d. The peaks corresponding to Fe, C, O, S, and Li are shown in Supplementary Fig. 19. In the high-resolution Fe 2p spectrum (Fig. 3c), the peaks corresponding to the Fe$^{3+}$ 2p$_{3/2}$, Fe$^{2+}$ 2p$_{1/2}$, Fe$^{3+}$ 2p$_{3/2}$, and Fe$^{2+}$ 2p$_{3/2}$ states were located at 726.2 eV, 724.2 eV, 713.1 eV, and 711.1 eV, respectively. All of the Fe 2p peaks moved to lower binding energies than those of SP-Fe$_3$O$_4$-C/S before cycling. These negative shifts suggested chemical interactions between the Fe-O species and S$_x^{2-}$, which led to an increase in the electron densities of the Fe species[49,69]. The high conductivity of Fe$_3$O$_4$ facilitated charge transfer, which enhanced catalytic conversion of the LiPSs. In addition, a new Fe-S signal at 707.9 eV was observed (Fig. 3c), indicating that the strong interaction between Fe$_3$O$_4$ and LiPSs led to the generation of FeS$_x$ species[70]. In the resolution S 2p spectrum (Fig. 3Fd), the peaks at 165.4 eV and 164.3 eV were ascribed to the S 2p$_{1/2}$ and 2p$_{3/2}$ states, respectively[71]. The peaks at 170.0 eV and 168.7 eV were assigned to thiosulfate[72]. Notably, a new peak appeared at 162.8 eV, which confirmed the formation of Fe-S bonds[70]. This result was consistent with the Fe 2p spectrum. The XPS results demonstrated chemical bonding between Fe$_3$O$_4$ and LiPSs, and the formation of Fe-S bonds led to a reduced energy barrier for LiPSs conversion[70,73].

Generally, uncontrolled deposition of ion- and electron-insulating Li$_2$S on a cathode surface hinders continuous LiPSs conversion and significantly slows the kinetics of LiPSs conversion[74–77]. Therefore, to demonstrate the fast catalytic conversion of LiPSs induced by the SP-Fe$_3$O$_4$-C, a Li$_2$S precipitation experiment was performed. The highest area capacity (245 mAh g$_s^{-1}$) and earliest current response (2.75 × 10$^3$ s) were observed for SP-Fe$_3$O$_4$-C (Fig. 3e). In comparison, the area capacities of B-Fe$_3$O$_4$-C and SP-C were 125 mAh g$_s^{-1}$ and 75 mAh g$_s^{-1}$, respectively (Fig. 3f, g). The current responses of B-Fe$_3$O$_4$-C and SP-C were 5.22 × 10$^3$ s and 13.39 × 10$^3$ s, respectively, which were much slower than that of SP-Fe$_3$O$_4$-C. Therefore, SP-Fe$_3$O$_4$-C showed the largest Li$_2$S deposition capacity and fastest nucleation/growth of Li$_2$S. These results reflected three facts: (1) compared with SP-C, SP-Fe$_3$O$_4$-C enabled faster catalytic conversion kinetics between the LiPSs and Li$_2$S due to the presence of Fe$_3$O$_4$; (2) the fastest current response of SP-Fe$_3$O$_4$-C indicated the rapid growth of Li$_2$S; and (3) the highest Li$_2$S deposition capacity of SP-Fe$_3$O$_4$-C suggested the conversion of more LiPSs to Li$_2$S. The entire process of LiPSs capture and catalytic conversion of the SP-Fe$_3$O$_4$-C cathode is illustrated in Fig. 3h.

**Study on the effect of the bicontinuous structure on the performance of Li-S batteries**

The advantages of the bicontinuous structure for mass/electron transfer of SP-Fe$_3$O$_4$-C/S are illustrated in Fig. 4a. First, the enhanced reaction kinetics were revealed by cyclic voltammetry (CV) of SP-Fe$_3$O$_4$-C/S, and the voltammograms were recorded over a voltage window of 1.7 to 2.8 V. In the contour maps for the CV curves (0.1–0.5 mV s$^{-1}$; Fig. 4b, c and Supplementary Fig. 20), the CV curves displayed two redox peaks: peak A at ~ 2.0 V corresponded to the reduction of S$_8$ to Li$_2$S$_x$ (X = 4, 6, 8), and peak B at ~ 2.3 V was ascribed to the subsequent transition to Li$_2$S$_2$/Li$_2$S[78]. Another oxidation peak (anode peak C at ~ 2.4 V) was assigned to the oxidation of Li$_2$S$_2$/Li$_2$S to S$_8$[78]. Importantly, compared with the B-Fe$_3$O$_4$-C/S-based battery, the SP-Fe$_3$O$_4$/C/S-based battery showed higher currents, lower polarization voltages and narrower half-peak widths (0.17 V) at each scanning rate, indicating faster redox reactions kinetics of LiPSs conversion.

Second, the mass transfer kinetics in the 3D continuous mesochannels were quantitated with the Li$^+$ diffusion coefficient ($D_{Li}^+$). The $D_{Li}^+$ was calculated via the Randles−Sevcik equation[79–81]:

$$I_p = 2.69 \times 10^5 n^{3/2} A D_{Li}^{1/2} v^{1/2} C_{Li} \qquad (1)$$

where $I_p$ is the peak current, $n$ is the number of electrons in the reaction ($n = 2$), $A$ is the electrode area ($A = 1.2$ cm$^2$), $v$ is the scanning rate (V/s), and $C_{Li^+}$ is the Li-ion concentration in the electrolyte ($C_{Li^+} = 1 \times 10^{-3}$ mol ml$^{-1}$)[18]. Based on the linear relationship between $I_p$ and $v^{1/2}$ for the main redox peaks (A, B and C in the CV curve), the $D_{Li}^+$ values for SP-Fe$_3$O$_4$-C/S were determined to be 1.83 × 10$^{-8}$, 8.12 × 10$^{-9}$ and 4.74 × 10$^{-8}$ cm$^2$ s$^{-1}$, respectively (Fig. 4d, e). In contrast, the $D_{Li}^+$ values of B-Fe$_3$O$_4$-C/S were 5.39 × 10$^{-9}$, 5.10 × 10$^{-9}$ and 4.18 × 10$^{-8}$ cm$^2$ s$^{-1}$, respectively. The SP-Fe$_3$O$_4$-C/S cathode had a higher $D_{Li}^+$ at each redox peak, indicating that electrolyte diffusion in SP-Fe$_3$O$_4$-C/S was better than that in B-Fe$_3$O$_4$-C/S (Supplementary Table. 4), which benefited from the 3D continuous mesochannels and contributed to the improved redox kinetics. Third, we used time-of-flight secondary-ion mass spectrometry (TOF-SIMS) to reveal the depth profiles of S$^-$ secondary ions in the cycled cathodes (0.2 C after 50 cycles) (Fig. 4f, g). With respect to the cycled B-Fe$_3$O$_4$-C/S cathode, the intensity of S$^-$ decreased with increasing etching depth, indicating nonuniform distribution of the S$^-$ on the outer surface of the cycled cathode. In contrast, the S$^-$ intensity in the SP-Fe$_3$O$_4$-C/S cathode was almost constant with increasing etching depth, and there was no obvious aggregation of S$^-$ on the cathode surface, indicating inhibition of LiPSs migration and a uniform distribution of S$^-$ in the whole SP-Fe$_3$O$_4$-C/S cathode, as illustrated in Fig. 4h. Fourth, the charge transfer capability of the SP-Fe$_3$O$_4$-C/S cathode was evaluated with in situ electrochemical impedance spectroscopy (EIS) during the 1st charge/discharge cycle (Fig. 4i). The results showed that the charge transfer resistance ($R_{ct}$) of the SP-Fe$_3$O$_4$-C/S cathode was significantly lower than that of the B-Fe$_3$O$_4$-C/S cathode. Upon recharging, no significant changes in the charge transfer resistance were observed, suggesting stable solid−liquid−solid conversion kinetics[78]. Notably, the $R_{ct}$ of the SP-Fe$_3$O$_4$-C/S cathode was always lower than that of the B-Fe$_3$O$_4$-C/S cathode during the entire discharge/charge process, verifying that fast electron transfer occurred in the 3D continuous carbon framework of SP-Fe$_3$O$_4$-C/S (Fig. 4i).

After evaluating the role of the bicontinuous structure, the LiPSs conversion mechanism was also studied. In situ Raman spectroscopy was used to monitor the redox reaction, in which a small circular hole was created on the side of the cathode, which exposed the SP-Fe$_3$O$_4$-C/S cathode to the incident laser beam (Supplementary Fig. 21). At the open circuit voltage (OCV), three predominant peaks appeared at 153.5, 219.4, and 472.9 cm$^{-1}$ and were attributed to solid S$_8$, and they gradually decreased after discharging to 2.26 V. Afterward, the signals for the S$_6^{2-}$ and S$_4^{2-}$ + S$_5^{2-}$ species appeared at 406.7 and 461.1 cm$^{-1}$, respectively, and became more intense as the discharge time increased. Then, the intensities of all peaks gradually became weaker until the end of discharge. Upon charging to 2.24 V, the signals for the sulfur-containing species reappeared. After charging to 2.8 V, the polysulfide signals had almost disappeared, and S$_8$ was regenerated. During the whole charge/discharge process, signals attributed to the discharge product Li$_2$S were not found owing to interference from the solvent and poor crystallinity[82]. The opposite reaction processes were detected during discharging/charging, demonstrating good reversibility for the conversion of LiPSs in the SP-Fe$_3$O$_4$-C/S cathode.

Finally, the finite element method based on COMSOL Multiphysics software was used for simulations of the bicontinuous structure in the SP-Fe$_3$O$_4$-C/S cathode. A model with continuous framework and channel was constructed and then the channels were filled with sulfur to simulate the SP-Fe$_3$O$_4$-C/S cathode. In contrast, another model with discontinuous mesopores was also constructed and then

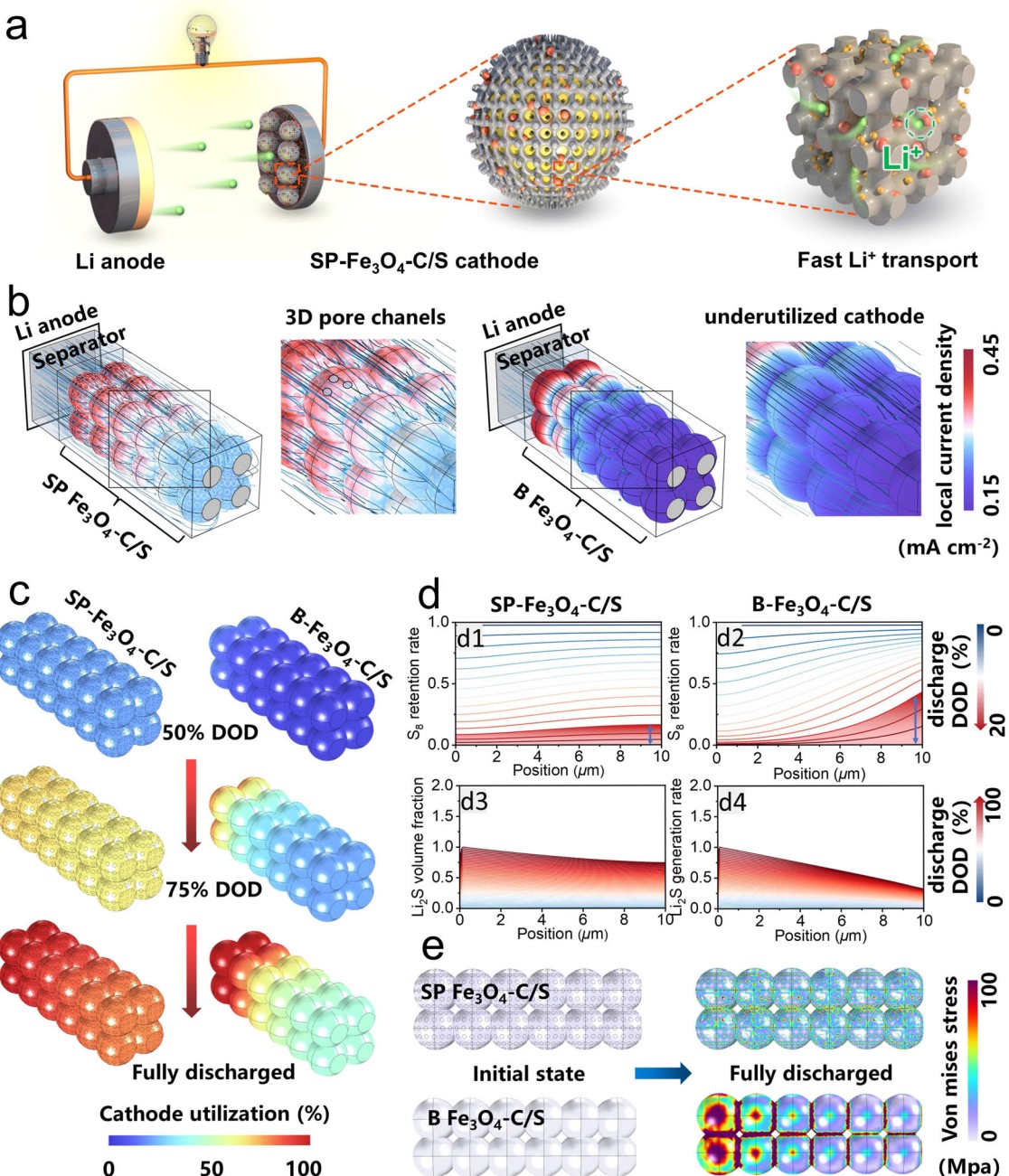

**Fig. 5 | Finite element simulation. a** Schematic illustration of the SP-Fe$_3$O$_4$-C/S-based Li-S battery and Li$^+$ diffusion in the SP-Fe$_3$O$_4$-C/S cathode; carbon frame, Fe$_3$O$_4$ nanoparticles, sulfur, and lithium ion are marked with grain, red, yellow, and green color, respectively. **b**–**e** Finite element analysis of the electrochemical and mechanical performance: **b** Local current density distributions for the Li$_2$S-generated reaction in the SP-Fe$_3$O$_4$-C/S and B-Fe$_3$O$_4$-C/S models; **c** Simulation of the cathode utilization evolution in the SP-Fe$_3$O$_4$-C/S and B-Fe$_3$O$_4$-C/S models from 50% depth of charge (DOD) to 100% DOD, where cathode utilization was defined as the ratio of the actually generated Li$_2$S to the theoretically generated Li$_2$S; **d** S$_8$ retention ratio evolution from 0 to 20% DOD: SP-Fe$_3$O$_4$-C/S (d1) and B-Fe$_3$O$_4$-C/S (d2); Li$_2$S generation ratio evolution from 0 to 20% DOD: SP-Fe$_3$O$_4$-C/S (d3) and B-Fe$_3$O$_4$-C/S (d4); **e** Simulated electrode expansion stress after S$_8$ was converted into Li$_2$S in the SP-Fe$_3$O$_4$-C/S and B-Fe$_3$O$_4$-C/S models.

filled with sulfur to simulate the B-Fe$_3$O$_4$-C/S cathode. The Li$^+$ diffusion model within the SP-Fe$_3$O$_4$-C/S cathode is presented in Fig. 5a. The simulation results (Fig. 5b) showed that the interpenetrating mesopores increased the reaction interfacial area for the sulfur redox reactions and enabled a more homogeneous local current distribution, leading to improved reaction kinetics for the whole cathode. In comparison, due to the difficulty of ion transport, the conversion of polysulfides in areas distant from the separator within the B-Fe$_3$O$_4$-C/S cathode occurred at slower rates and exhibited uneven current distributions, leading to low sulfur utilization throughout the entire cathode. This contrast was particularly evident in thick electrodes with high sulfur loadings, as supported by the finite element method simulations (Fig. 5c). As shown in Fig. 5c, the sulfur in the SP-Fe$_3$O$_4$-C/S cathode was almost fully utilized throughout the whole electrode during the discharge process. Moreover, the quantitative distributions of S$_8$ and Li$_2$S in the cathode after discharge are given in Fig. 5d. There was faster S$_8$ conversion in the SP-Fe$_3$O$_4$-C/S cathode during the discharge process than in the B-Fe$_3$O$_4$-C/S cathode (Fig. 5d1, 2). Moreover,

the $Li_2S$ generation ratio was higher than that in the B-$Fe_3O_4$-C/S cathode after full discharge, and $Li_2S$ was uniformly deposited throughout the entire SP-$Fe_3O_4$-C/S cathode (Fig. 5d3, 4). This may have led to a higher sulfur utilization efficiency in accordance with the discharge curves with higher capacities and lower polarization voltages for SP-$Fe_3O_4$-C/S relative to B-$Fe_3O_4$-C/S (Supplementary Fig. 22). Another factor to be considered is the mechanical stability of the cathode during the discharge process, as there can be an 80% volume expansion in converting $S_8$ to $Li_2S$. A large volume expansion may lead to severe mechanical stress generated by mutual compression among active particles. The simulation results (Fig. 5e) showed that inhomogeneous sulfur utilization resulted in a nonuniform distribution of the mechanical stress in B-$Fe_3O_4$-C/S. In contrast, evenly distributed $Li_2S$ deposition in SP-$Fe_3O_4$-C/S led to a homogeneous volume expansion and a smaller stress gradient distributed across the entire electrode. Simultaneously, the bicontinuous structure accommodated the deposition of $Li_2S$, which reduced the volume expansion and enabled obviously weakened extrusion of sulfur particles, as shown in Fig. 5e, which mitigated the stress concentration. A homogeneous stress distribution may have prevented damage and collapse of the electrode materials due to mechanical failure, which would be highly beneficial for long-term cycling with high sulfur loadings. In a word, the finite element simulations confirmed that SP-$Fe_3O_4$-C/S had rapid ion transport capability, fast $S_8$ reduction kinetics, and structural advantages in tolerating large volume changes.

In summary, $Fe_3O_4$-dopped carbon cubosomes with a single primitive bicontinuous structure were constructed from EGCG and $Fe^{3+}$ precursors with the help of a BCP-based cubosome template. The obtained SP-$Fe_3O_4$-C exhibited 3D ordered interpenetrating mesochannels and a 3D continuous carbon skeleton, high sulfur loading and facilitated electron/mass transport within the cathode. Moreover, the uniformly distributed $Fe_3O_4$ on the continuous carbon skeleton adsorbed the LiPSs and catalyzed their conversion. Impressively, the SP-$Fe_3O_4$-C-based cathode in the Li-S battery provided a high reversible capacity of 1303.4 mAh $g^{-1}$ at 0.2 C, a high rate capability of 691.8 mAh $g^{-1}$ at 5 C and an impressive cycling life over 1200 cycles at 1 C with a low decay rate of 0.027% per cycle. In addition, computational simulations demonstrated that the unique bicontinuous channels and frameworks achieved high sulfur utilization, uniform $Li_2S$ deposition and fast $Li^+/e^-$ transportation with SP-$Fe_3O_4$-C. This work provides a promising example of a carbon-based sulfur host with a unique structure for use in high-performance Li-S batteries. We also believe that the preparation strategy and mechanism presented here will contribute to future construction of multifunctional cathode materials for high-performance energy storage devices.

## Methods

### Chemicals and materials

Poly(ehylene glycol) methyl ether ($M_n$ = 2000 g $mol^{-1}$), copper(I) bromide (CuBr), styrene (St), 2-bromoisobutyryl, triethylamine ($Et_3M$), $N,N,N',N',N''$-pentamethyldiethylenetriamine (PMDETA), tetrahydrofuran (THF), alcohol, 1,4-dioxane, $N,N$-dimethylformamide (DMF), (-)-epigallocatechin gallate (EGCG), iron(III) chloride hexahydrate ($FeCl_3·6H_2O$), potassium hydroxide (KOH), 3-($N$-morpholino)propanesulfonic acid (MOPs), 1,3-dioxolane (DOL), dimethoxyethane (DME), polyvinylidene fluoride (PVDF) and 1-methyl-2-pyrrolidinone (NMP) were purchased from Adamas Reagent (shanghai) and Energy Chemical. Deionized water was used in all steps. St was distilled before using. Other agents were used without purification.

### The synthesis of PS-$b$-PEO$_{45}$

The PS-$b$-PEO$_{45}$ block copolymer was synthesized by atom transfer radical polymerization (ATRP) method[83]. Firstly, PEO-Br macroinitiator was synthesized as follows: (1) PEO2000 (20 g) was dissolved in freshly distilled dichloromethane (DCM, 100 mL) in a Schlenk-flask under stirring (500 rpm). After the addition of $Et_3N$ (3.5 mL), the mixture was cooled to 0 °C and kept stirring for 1 h. (2) Then, 2-bromoisobutyryl bromide (7.7 mL) was added dropwise under stirring. The reaction mixture was stirred for 12 h in an ice-water bath. (3) The solution after the reaction was recrystallized by isopropanol for several times until the solid turned white. (4) Then, the white solid was fully dissolved by DCM (20 mL) and the almost colorless solution was obtained. Subsequently, the solution was added dropwise into 500 mL cold ether under stirring to precipitate the product. The generated white precipitate was filtered and washed with cold ether for three times. (5) The obtained snow-white product (PEO-Br) was dried under vacuum at room temperature overnight. Second, the PS-$b$-PEO$_{45}$ block copolymer was synthesized through the following steps: (1) 1 g prepared PEO$_{45}$-Br, 50 mg CuBr and 80 $\mu$L PMDETA were added into a Schlenk flask. Afterwards, a calculated amount of styrene (9 mL for PS$_{241}$-$b$-PEO$_{45}$ and 5 mL for PS$_{180}$-$b$-PEO$_{45}$) was transferred to the flask. (2) The mixture in the flask was mixed by ultrasonic and further deoxygenated by three freeze-pump-thaw cycles. (3) The flask was subsequently immersed in an oil bath at 120 °C and reacted for 6 h. (4) Then, the product was dissolved in DCM (50 mL) and then passed through a short column of basic alumina to remove copper complexes. (5) The mixture solution was concentrated and then dropped into a large amount of methanol (500 mL) under stirring to precipitate the white precipitate. (6) The final product (PS-$b$-PEO$_{45}$) was filtered and dried under vacuum at room temperature for 24 h.

### Preparation of the PC template

DP-PCs were prepared by a cosolvent method[50]. Typically, 20 mg PS$_{241}$-$b$-PEO$_{45}$ was dissolved in 2 mL mixture solution of dioxane/dimethylformamide (92:8, v/v). The solution was stirred for 4 h at room temperature for complete dissolution of the copolymer. Then 2 mL water was added at a controlled rate (1 mL $h^{-1}$) to the polymer solution under stirring (200 rpm). The mixtures were dialyzed against water to remove organic solvent and freeze the morphology. Then, DP-PCs were obtained by centrifugation at 955 × $g$ for 3 min.

### Preparation of SP-$Fe_3O_4$-C

Aqueous solution of EGCG (30 mM, 1 mL) and $FeCl_3·6H_2O$ (30 mM, 1.3 mL) solution were sequentially added to the DP-PC aqueous solution (1 mg $mL^{-1}$, 20 mL). The suspension was vigorously stirred at 500 rpm for 2 h to ensure the complete diffusion of the EGCG and $Fe^{3+}$ into the pore channels of DP-PC templates. The pH of the suspension was then raised by adding 25 mL of 3-($N$-morpholino)propanesulfonic acid (MOPs) buffer (100 mM) solution (pH = 6.8−7.2). The reaction was continuously stirred for another 20 min. The product was collected and purified by centrifugation at 955 × $g$ for 3 min and washed with ethanol and deionized water for at least three cycles. Finally, the collected product was dried at 80 °C for 12 h to yield MPN@DP-PC composite. Pyrolysis of the MPN@DP-PC was carried out in a tubular furnace under nitrogen atmosphere at 350 °C for 2 h and 800 °C for 2 h, respectively, with a heating rate of 2 °C $min^{-1}$.

### Control experiments on the effect of the mole ratio of EGCG and $Fe^{3+}$ on the formation of SP-$Fe_3O_4$-C

The preparation conditions were similar to those described in section 'Preparation of SP-$Fe_3O_4$-C' except that the molar ratios of EGCG and $Fe^{3+}$ were varied. In these control experiments, the molar ratio (it was also equal to the volume ratio since the precursor concentration was the same, [EGCG] = [$Fe^{3+}$] = 30 mM) of EGCG and $Fe^{3+}$ was adjusted to 1:0.5, 1:0.9, 1:1, 1:1.3, 1:1.7, and 1:2.1, respectively.

### Preparation of B-$Fe_3O_4$-C

Firstly, PS$_{180}$-$b$-PEO$_{45}$ was prepared by ATRP as described in section 'The synthesis of PS-$b$-PEO$_{45}$'. Afterwards, 20 mg PS$_{180}$-$b$-PEO$_{45}$ was

dissolved in 4 mL THF to prepare the polymer solution, and quickly pour the mixed solution of 4 mL ethanol and 8 mL water into the polymer solution under vigorous stirring to prepare the spherical micelle solution. Then, 0.45 mmol EGCG and 0.59 mmol $Fe^{3+}$ were added sequentially to the above micelle solution. After stirring for another 2 h, an equal volume of pH buffer (prepared as described in section Preparation of $SP\text{-}Fe_3O_4\text{-}C$) was added. The mixture solutions were stirred for 20 min before purification. The desired $B\text{-}Fe_3O_4\text{-}C$ was obtained through a pyrolyzing process (post-treated details are given in section Preparation of $SP\text{-}Fe_3O_4\text{-}C$).

## Preparation of $SP\text{-}Fe_3O_4\text{-}C/S$ and $B\text{-}Fe_3O_4\text{-}C/S$

Impregnation of sulfur was carried out by a melt-diffusion method[84]. Briefly, $SP\text{-}Fe_3O_4\text{-}C$ and sulfur were well-mixed as a mass ratio of 1:3. Then the mixture was transferred into a quartz tube furnace and heated at 155 °C for 12 h, resulting in the $SP\text{-}Fe_3O_4\text{-}C/S$ sample. $B\text{-}Fe_3O_4\text{-}C/S$ was prepared by the same method under similar conditions.

## Adsorption test of lithium polysulfides[85]

The $Li_2S_6$ solution was prepared by dissolving sulfur and $Li_2S$ (5:1, molar ration) in a DOL/DME (1:1, v/v) solution at 60 °C for 24 h. All samples ($SP\text{-}Fe_3O_4\text{-}C$, $B\text{-}Fe_3O_4\text{-}C$, and carbon black) were dried at 80 °C under vacuum oven for 12 h before the absorption test. Then, 10 mg samples were added into the $Li_2S_6$ solution (4 mM, 5 mL) and kept for 24 h in an argon-filled glove box. The supernatants were taken for the UV-vis absorption spectroscopy analysis.

## $Li_2S$ precipitation experiments[86]

The $Li_2S_8$ solution was prepared by uniformly mixing $Li_2S$ and S at a mole ratio of 1:7 in the electrolyte, which consisted of 1 M LiTFSI salt (DOL/DME 1:1, v/v) with 1 wt% $LiNO_3$. The batteries for the $Li_2S$ precipitation experiments were assembled by using lithium foil as the counter electrode, $SP\text{-}Fe_3O_4\text{-}C$ coated on carbon cloth as the working electrode and 20 μL $Li_2S_8$ solution was dropping onto the working electrode. The cells were galvanostatically discharged to 2.06 V at 0.112 mA, then maintained at 2.05 V until the current was less than 0.01 mA. The procedures for the $Li_2S$ precipitation experiments on $B\text{-}Fe_3O_4\text{-}C$ and SP-C were similar except the use of different carbon samples.

## Electrochemical measurements

The electrodes for the Li-S cells were prepared by mixing the active material ($SP\text{-}Fe_3O_4\text{-}C/S$ or $B\text{-}Fe_3O_4\text{-}C/S$), carbon black and PVDF in the mass ratio of 7:2:1 in NMP. The mixture was coated on an aluminium (Al) foil collector, followed by drying at 60 °C in a vacuum oven for 12 h. The mass of active materials in the cathode was measured by the mass difference between the assembled cathode and the pure Al foil collector. The Li-S cells were assembled with lithium metal foil as the anode, polypropylene (PP) membrane served as the separator, and 50 μL electrolyte in each cell. The entire process of battery assembly was carried out in an Ar-filled glove box. The electrochemical performance was tested on a LAND Battery Tester with a voltage window of 1.7 to 2.8 V. Cyclic voltammetry and in situ impedance testing were performed on a bio-logical electrochemical workstation with the voltage range from 1.7 to 2.8 V. Each electrochemical datum was the average value based on the measurements of three cell samples. All the electrochemical tests were carried out in a constant temperature laboratory (25 °C). All the specific capacities of the cells have been normalized based on the weight of sulfur.

## Characterizations

Liquid-phase nuclear magnetic resonance ($^1H$ NMR): $^1H$ NMR analyses were performed on a Bruker 400 (400 MHz for proton) spectrometer at room temperature with 16 scans. Gel permeation chromatography (GPC): Gel permeation chromatograph was performed at on a EcoSEC-HLC-8321GPC/HT from Tosoh, Japan. The test temperature was 40 °C,

the standard sample was monodispersed styrene, and the mobile phase was tetrahydrofuran. The sample was prepared by dissolving 2 mg of polymer sample in HPLC tetrahydrofuran, filtered through a filter and injected into a GPC test vial. Scanning electron microscopy (SEM): Scanning electron microscopy was performed on a JSM-7800F super-resolution field emission scanning electron microscope from Japan Electronics Corporation, with an accelerating voltage of 5.0 kV and a resolution of 1.2 nm. A drop of the sample dispersion was added to a clean silicon wafer during sample preparation and allowed to evaporate for 24 h. The wafer was washed in advance with deionized water, acetone, and anhydrous ethanol, then immersed in ethanol and removed from the ethanol solution before use, and allowed to dry naturally. The sample can be added only after natural drying. Transmission electron microscopy (TEM): Transmission electron microscopy was performed using a Tecnai G2 spirit Biotwin biological transmission electron microscope (FEI, USA), with an accelerating voltage of 120 kV and a point resolution of 0.49 nm. The samples were prepared by placing a drop of sample dispersion on a copper grid and evaporating for 24 h. Magnetic samples were prepared by double networking. Sections were prepared by embedding the samples with resin and then slicing them with an EM/UC7 ultra-thin sectioning machine from Leica, Germany. The samples were sliced using a UC7 ultrathin slicer with a thickness of 100 nm. Nitrogen adsorption-desorption measurement: Nitrogen adsorption isotherms were measured at 77 K on an Autosorb-iQA3200-4 sorption analyzer (Quantatech, USA). The samples were degassed under vacuum at 393 K for 8 h before testing. The Brunauer−Emmett−Teller (BET) method was used to calculate the specific surface area. The desorption branch of the isotherm and Barrett-Joyner-Halenda (BJH) method was used to calculate the pore size distribution and pore volume. Fourier transform infrared (FTIR) spectroscopy: FTIR spectra were recorded on a Nicolet 6700 infrared spectrometer from Thermo Fisher Scientific (USA). The sample was dried under vacuum at 80 °C for 8 h. A small amount of samples powder was taken and fully ground with dried KBr powder, and then pressed into thin slices. X-ray photoelectron spectroscopy (XPS): XPS was characterized using an AXIS Ultra DLD-type X-ray photoelectron spectrometer (Shimadzu, Japan) equipped with the monochromatic Kα radiation of the Al target as the X-ray source (1486.6 eV) and the binding energy of the C 1s peak (284.6 eV). X-ray diffraction (XRD): XRD patterns were recorded on a D8 DaVinci X-ray diffractometer (Bruker, Germany) with a test angle range of 5−80° (2θ) and a rate of 6°/min. Small angle X-ray scattering (SAXS): Small angle X-ray scattering was performed at the BL10U1 small-angle X-ray scattering station (third generation synchrotron light source) with an X-ray wavelength (λ) of 2 Å and a tube length of 27.6 m using the Shanghai Synchrotron Radiation Source (SSRF). Thermogravimetric analysis (TGA): Thermogravimetric analysis (TGA) is carried out on a TA SDT Q650 (USA). To acquire the Fe content. The samples were heated to 800 °C under air atmosphere with a heating rate of 10 °C/min. While to obtain the sulfur contents, the sulfur encapsulated samples were heated to 400 °C under high-purity nitrogen with a heating rate of 10 °C/min. Raman Spectroscopy: Raman spectroscopy was conducted on an In Via Qontor confocal micro-Raman spectrometer (Renishaw, UK) to characterize the graphitic degree of carbon materials. In situ Raman: In situ Raman spectra was obtained using a self-made cell, the schematic diagram of the in situ cell device is shown in Supplementary Fig. 21. Raman measurements were acquired using a Horiba LabRAM HR Evolution at 532 nm. Ultraviolet−visible spectrophotometer (UV-vis): UV-vis spectrophotometer (UV-2600, Shimadzu, Japan) was used for the UV test of LiPSs adsorption experiment. Time-of-Flight Secondary Ion Mass Spectrometry (TOF-SIMS): TOF-SIMS characterization was carried out on the ION TOF SIMS 5−100 instrument. For the TOF-SIMS 3D analysis, a 3 k eV $Cs^+$ beam with a current of 30 nA was used on the electrode. The imaged area was 40 μm × 40 μm. The cathodes were galvanostatic cycled within the voltage window of 1.7 to 2.8 V at 0.2 C

for 50 cycles before the TOF-SIMS characterization. The samples for TOF-SIMS measurements were prepared and sealed in bags in a glove box filled with argon. After transferring the samples encapsulated in valve bags filled with argon, the TOF-SIMS measurements were conducted under vacuum.

## Data availability

The data that support the findings of this study are available in the main text and Supplementary Information. Should any raw data files be needed in another format, all of them available from the corresponding authors upon reasonable request.

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

## Acknowledgements

The authors appreciate the financial support from National Natural Science Foundation of China (22225501 and 52073173, Y.M.; 52203268, F.X.; 52072205, G.Z.), Open Fund of Shanghai Jiao Tong University-Shaoxing Institute of New Energy and molecular Engineering (JDSX2022025, Y.M.), Program of Shanghai Academic/Technology Research Leader (23XD1431700, C.Z.), Shanghai Engineering Research Center of Specialized Polymer Materials for Aerospace (18DZ2253500, C.Z.), the Joint Funds of the National Natural Science Foundation of China (U21A20174, C.Z.), the National Key Research and Development Program of China (2019YFA0705703, G.Z.), the Joint Funds of the National Natural Science Foundation of China (U21A20174, G.Z.), Guangdong Innovative and Entrepreneurial Research Team Program (2021ZT09L197, G.Z.), Shenzhen Science and Technology Program (KQTD20210811090112002, G.Z.), and the Overseas Research Cooperation Fund of Tsinghua Shenzhen International Graduate School (the fund was provided by the school platform and no specific fund number, G.Z). They also appreciate Dr. Feng Tian and the time-resolved USAXS Beamline BL10U1 at Shanghai Synchrotron Radiation Facility (SSRF) for SAXS measurements.

## Author contributions

H.L., F.X., G.Z., and Y.M. conceived and designed the project. H.Z. performed most experiments and interpreted the data. H.L. conducted the electrochemical measurements and analyzed the data. R.L. performed the related synthesis of MPN. M.Z. contributed to the finite element simulation. Y.J. contributed to the DFT calculation. L.X. helped with the electrochemical data analysis. C.Z. and T.H. contributed to helpful discussions. X.W. and Z.P. helped to revise and polish the article. H.Z., H.L., F.X. G.Z. and Y.M. co-wrote the manuscript with contributions from all co-authors. All authors have given approval to the final version of the manuscript.

## Competing interests

The authors declare no competing interests.
