## [Peer Review File · Nature Communications]

Fe₃O₄-doped mesoporous carbon cathode with a plumber's nightmare structure for high-performance Li-S batteriesREVIEWER COMMENTS

Reviewer #1 (Remarks to the Author):

This manuscript reports the preparation of Fe₃O₄-doped carbon cubosomes with an ordered single-network “plumber's nightmare” structure (SP-Fe₃O₄-C). The design strategy is interesting and seems innovative to the best knowledge of the reviewer. The Li-S battery performance is not the best, but comparable with the state-of-the-art. Overall, the manuscript is well prepared. It is recommended for publication after addressing the following comments.

1. TOF-SIMS was used to reveal the depth profiles of S- secondary ions in the cycled cathodes, which well demonstrated the strong capability to inhibit the polysulfides shuttling. However, how did the authors prepare the sample for the test, as Li₂S is easy to oxidize when exposed to the air?
2. The long-cycling performance of SP-Fe₃O₄-C is very impressive. Although the charge/discharge is very reversible, CE cannot be 100%, please double check and make it reasonable.
3. Fe₃O₄ particles are very important for adsorption of LiPSs. In the calculation section, why choose the (220) plane of Fe₃O₄ for studying the adsorption of LiPSs?
4. In Fig. (4b,4c), it is a good example to make contour maps of CV, which can easily see some important information (current/ polarization). I am curious what the half-peak width means?
5. The authors simulated the stress distribution of the electrode in Figure 5e. Could the authors elaborate on the specific methodology used to achieve this simulation? In addition, can the authors explain why the mesoporous channels contribute to mitigating the stress concentration?
6. Some expressions in the manuscript are misleading or incorrect. For example, “lower capacity decay of only 0.024% after 120 cycles”, “per cycle” should be added. The same problem can also be found in the next sentence. In Page 16, “These negative shifts suggested chemical interactions between the Fe-O species and Sx²⁻, which led to increases in the electron densities of the Fe species”. In this sentence, “increases” should be “an increase”. In the SI, the unit of the vertical axis should be mA but not A.

Reviewer #2 (Remarks to the Author):

In this manuscript, Mai and coworkers reported a novel type of porous carbon particles with 3D continuous mesochannels and polar species as the cathode material for high-performance Li-S batteries. In the preparation process, a removable polymeric colloid template (polymer cubosomes), a biomass-derived Fe metal-phenolic network (Fe-MPN), and a programmed pyrolysis were employed. The resultant SP-Fe₃O₄-C possessed continuous open channels, interconnected carbon frameworks and uniformly distributed Fe₃O₄ particles, which makes them promising sulfur host materials with high sulfur loading, strong LiPS capture capability, smooth mass/electron transport, as well as fast catalytic conversion of the sulfur species. Thus, SP-Fe₃O₄-C based Li-S batteries delivered an impressive comprehensive performance. In a word, this study provides a promising structure for improving the performance of Li-S batteries, and cast a new light on the designing of multifunctional electrode materials for high-performance energy storage devices. The manuscript was well written and the data are convincing. I would like to recommend the publication of this work after some minor revisions.

1. In order to present a comprehensive comparison of the changes in the samples before

and after the sulfur loading, the pore volumes of SP-Fe₃O₄-C, SP-Fe₃O₄-C/S, B-Fe₃O₄-C and B-Fe₃O₄-C/S should be given in the manuscript, and their data should be marked in the N₂ adsorption–desorption isotherms (Fig. S10g and Fig. S13d).

2. The SEM or TEM images of the bicontinuous porous carbon cubosomes look very beautiful and pure. I am wondering how to purify the cubosomes during the preparation.
3. Page 7 in the manuscript, the sentence of “The SP-Fe₃O₄-C particles were obtained by pyrolysis of the resultant MPN@DP-PCs at 350 °C and 800 °C for 2 h” is a little ambiguous. Was the pyrolysis performed at each temperature for 2h? or totally for 2h?
4. Page 20, in the in-situ EIS test, the charge transfer resistance of the SP-Fe₃O₄-C/S cathode was lower than that of the B-Fe₃O₄-C/S cathode, while the IG/ID values in their Raman spectra were almost the same. Please give reasonable explanations.
5. Page 22, in Fig. 5b, the model of SP-Fe₃O₄-C/S and B-Fe₃O₄-C/S looks like the same, except for the distribution of the colors. More detailed descriptions and annotations are needed to clarify the relationship between the models and the two samples employed in this study.

Reviewer #3 (Remarks to the Author):

This manuscript demonstrated a kind of Fe₃O₄-doped carbon material with single primitive bicontinuous topology as a host of sulfur cathode in Li-S batteries. The authors combined the experiments with finite element simulation to illustrate that this cathode shows good ion transport capability and fast kinetics. And this cathode also displays good cycle life over 1200 at 1C. However, there are some questions need to be further addressed.

1. In Figure 4d, we can see the linear fit of A2 peak is not good. Maybe the relationship between the peak current of A2 and scan rate do not follow the Randle-Sevcik equation.
2. The authors claimed that the channels of SP-Fe₃O₄-C were almost completely filled with sulfur. However, in Figure S10a-c, we cannot know whether the sulfur can be filled in the pores deep inside the structure. And in the BET test, if the pores of the outer surface of the structure are filled with sulfur completely but the inside pores not, the surface area will also be low due to the blocking of the sulfur on the surface. Please provide more data to confirm that the channels of SP-Fe₃O₄-C were almost completely filled with sulfur.
3. In page 16, the authors mentioned that the high conductivity of Fe₃O₄ facilitated charge transfer. What does high conductivity mean? Fe₃O₄ is only a kind of semiconductor.
4. There are some mistakes in this manuscript. In Page 23, Line 402, Li₂S should be corrected to Li₂S. The authors should check the whole manuscript carefully.
5. In Figure 4(i3), during discharge, we can see a new semicircle formation in the EIS plot. Please explain the reason.
6. Whether Fe₃O₄ shows some redox reactions in the operating voltage range of Li-S battery.

Response to the reviewers' comments:

Reviewer #1 (Remarks to the Author):

This manuscript reports the preparation of Fe₃O₄-doped carbon cubosomes with an ordered single-network “plumber's nightmare” structure (SP-Fe₃O₄-C). The design strategy is interesting and seems innovative to the best knowledge of the reviewer. The Li-S battery performance is not the best, but comparable with the state-of-the-art. Overall, the manuscript is well prepared. It is recommended for publication after addressing the following comments.

Response: We appreciate the reviewer for the positive comments on our work.

1. TOF-SIMS was used to reveal the depth profiles of S⁻ secondary ions in the cycled cathodes, which well demonstrated the strong capability to inhibit the polysulfides shuttling. However, how did the authors prepare the sample for the test, as Li₂S is easy to oxidize when exposed to the air?

Response: We appreciate this valuable comment. The samples for the TOF-SIMS measurements were prepared and sealed in valve bags in a glove box filled with argon. After transferring the samples encapsulated in valve bags filled with argon, the TOF-SIMS measurements were conducted under vacuum. Thus, Li₂S would not contact with the air during the sample preparation, transfer and measurement, which guaranteed the credibility of the TOF-SIMS results. We have added the details for the preparation of the TOF-SIMS samples in the experimental part of the revised manuscript (Page 31, Lines 564-566).

2. The long-cycling performance of SP-Fe₃O₄-C is very impressive. Although the charge/discharge is very reversible, CE cannot be 100%, please double check and make it reasonable.

Response: This is a very constructive suggestion, following which we checked the CE data carefully. The average CE value of SP-Fe₃O₄-C was calculated to be 99.94% (two decimal numbers were retained). We have revised the CE value in the revised manuscript (Page 13, Line 231).

3. Fe₃O₄ particles are very important for adsorption of LiPSs. In the calculation section, why choose the (220) plane of Fe₃O₄ for studying the adsorption of LiPSs?

Response: According to previous reports, all crystal planes of Fe₃O₄ can chemically adsorb

polysulfides (LiPSs) (e.g. *Adv. Energy Mater.* **2021**, *11*, 2100673; *ACS Nano* **2019**, *13*, 8986; *Small* **2023**, *19*, 2207924; *J. Mater. Chem. A* **2020**, *8*, 24117). In our study, the (220) and (311) planes were found to induce the main peaks in the XRD pattern of Fe₃O₄ in SP-Fe₃O₄-C (Supplementary Fig. 8a). Moreover, based on the HRTEM images of the Fe₃O₄ nanoparticles in SP-Fe₃O₄-C (Fig. 1f), the (220) plane of Fe₃O₄ was the most exposed crystal face. Furthermore, it has been reported that the (220) plane of Fe₃O₄ can induce shorter Fe-S bond and thus higher binding energy than those induced by other crystal planes, indicating the (220) plane might have higher chemical affinity with LiPSs (e.g. *Chem. Eng. J.* **2021**, *410*, 128153). Therefore, as a typical example, we chose the (220) plane for simulating the adsorption of LiPSs (Fig 3a).

4. In Fig. (4b,4c), it is a good example to make contour maps of CV, which can easily see some important information (current/polarization). I am curious what the half-peak width means?

Response: The half-peak width referred to the intermediate value of the charge/discharge voltage difference. The higher value of the half-peak width, the higher of the polarization is (*ACS Nano* **2021**, *15*, 7318; *Small* **2022**, *18*, 2104224). As seen in Figure 4b, the half-peak width of Peak A in SP-Fe₃O₄/C was only 0.17 V, smaller than that of B-Fe₃O₄/C (0.21 V), indicating a lower polarization and faster reaction kinetics in SP-Fe₃O₄/C.

5. The authors simulated the stress distribution of the electrode in Figure 5e. Could the authors elaborate on the specific methodology used to achieve this simulation? In addition, can the authors explain why the mesoporous channels contribute to mitigating the stress concentration?

Response: These are excellent questions. The specific simulation process can be divided into two parts. The first part encompasses the dissolution of S₈ and the deposition of Li₂S, involving solid-liquid transformation. Notably, there are big differences in the molar volume of S₈ and Li₂S, one mole of S₈ corresponds to eight moles of Li₂S, thereby giving rise to the disparities in the volume changes within the cathode. When determining the mass fraction of S₈ inside the electrode, the volume change of the entire electrode can be identified by calculating the conversion rate of S₈ and the generation rate of Li₂S during the simulation process. Subsequently, based on the calculated volume changes, displacements at the particle surfaces within the cathode can be derived, which are then utilized as the input parameters for the mechanical field. This allows for

the simulation of the stress variations in the charging process of the lithium-sulfur battery. Mesoporous channels play a dual role in mitigating the stress concentration. First, they facilitate rapid lithium-ion transport, promoting a uniform distribution of local current densities within the electrode. The homogeneous current density profile, in turn, engenders a more uniform volumetric change across the electrode. Second, the additional void space afforded by the mesopores accommodates the volume expansion, forestalling the interparticle compression, thereby further alleviating the stress localization.

6. Some expressions in the manuscript are misleading or incorrect. For example, “lower capacity decay of only 0.024% after 120 cycles”, “per cycle” should be added. The same problem can also be found in the next sentence. In Page 16, “These negative shifts suggested chemical interactions between the Fe-O species and S_x^{2-} , which led to increases in the electron densities of the Fe species”. In this sentence, “increases” should be “an increase”. In the SI, the unit of the vertical axis should be mA but not A.

Response: We appreciate the reviewer’s carefulness. We have carefully checked and revised these issues in the revised manuscript and the revised Supplementary Information (SI).

Reviewer #2 (Remarks to the Author):

In this manuscript, Mai and coworkers reported a novel type of porous carbon particles with 3D continuous mesochannels and polar species as the cathode material for high-performance Li-S batteries. In the preparation process, a removable polymeric colloid template (polymer cubosomes), a biomass-derived Fe metal-phenolic network (Fe-MPN), and a programmed pyrolysis were employed. The resultant SP-Fe₃O₄-C possessed continuous open channels, interconnected carbon frameworks and uniformly distributed Fe₃O₄ particles, which makes them promising sulfur host materials with high sulfur loading, strong LiPSs capture capability, smooth mass/electron transport, as well as fast catalytic conversion of the sulfur species. Thus, SP-Fe₃O₄-C based Li-S batteries delivered an impressive comprehensive performance. In a word, this study provides a promising structure for improving the performance of Li-S batteries, and cast a new light on the designing of multifunctional electrode materials for high-performance energy storage

devices. The manuscript was well written and the data are convincing. I would like to recommend the publication of this work after some minor revisions.

Response: We appreciate the reviewer for the positive comments on our work.

1. In order to present a comprehensive comparison of the changes in the samples before and after the sulfur loading, the pore volumes of SP-Fe₃O₄-C, SP-Fe₃O₄-C/S, B-Fe₃O₄-C and B-Fe₃O₄-C/S should be given in the manuscript, and their data should be marked in the N₂ adsorption-desorption isotherms (Fig. S10g and Fig. S13d).

Response: According to the reviewer's suggestions, we have added the structural parameters of SP-Fe₃O₄-C/S and B-Fe₃O₄-C/S in Supplementary Table. 1.

2. The SEM or TEM images of the bicontinuous porous carbon cubosomes look very beautiful and pure. I am wondering how to purify the cubosomes during the preparation.

Response: This is an excellent concern. The well-defined bicontinuous porous carbon cubosomes were obtained by pyrolysis of pure MPN@DP-PCs with good morphology. Thus, two key points should be noted. First, an optimized molar ratio of $n_{\text{EGCG}}/n_{\text{Fe}^{3+}} = 1.3$ (the two precursors of MPN) was employed to ensure the good morphology of MPN@DP-PCs. Second, taking advantage of the gravity difference of MPN@DP-PCs and the impurities of small sizes, pure MPN@DP-PCs could be collected by appropriate centrifugation at 3000 rpm for 3 min, and then washed with ethanol/deionized water (1/1, v/v) until the supernate was colorless.

3. Page 7 in the manuscript, the sentence of "The SP-Fe₃O₄-C particles were obtained by pyrolysis of the resultant MPN@DP-PCs at 350 °C and 800 °C for 2 h" is a little ambiguous. Was the pyrolysis performed at each temperature for 2 h? or totally for 2 h?

Response: We appreciate the reviewer's carefulness. Actually, the pyrolysis process of MPN@DP-PCs was conducted at each temperature for 2 h. We have revised the confused description in the revised manuscript (Page 7, Line 117; Page 26, Line 462).

4. Page 20, in the in-situ EIS test, the charge transfer resistance of the SP-Fe₃O₄-C/S cathode was lower than that of the B-Fe₃O₄-C/S cathode, while the I_G/I_D values in their Raman spectra were

almost the same. Please give reasonable explanations.

Response: The degree of graphitization of SP-Fe₃O₄-C and B-Fe₃O₄-C were almost the same ($I_G/I_D = 1.2$), indicating both of them possessed high conductivity. However, the electric conductivity of sulfur host materials is not the only decisive factor to determine their EIS resistance. In both SP-Fe₃O₄-C/S and B-Fe₃O₄-C/S cathodes, the sulfur loading content was 75wt%, and the mass percentage of the carbon host was 25wt%. Therefore, the uniformity of the distribution of insulated sulfur and lithium sulfide in the cathodes played an important role in the ionic/electronic conductivity of the whole electrodes. In the SP-Fe₃O₄-C/S cathode, the 3D continuous mesopores of SP-Fe₃O₄-C were conducive to the uniform distribution of sulfur and lithium sulfide, ensuring fast and smooth electron transfer throughout the entire cathode in the charge/discharge processes. While in the B-Fe₃O₄-C/S cathode, sulfur and lithium sulfide were mostly accumulated on the cathode surface during operation (Supplementary Fig. 16), which could increase the EIS impedance. Therefore, it is reasonable that the experimental results of *in situ* EIS showed that the SP-Fe₃O₄-C/S cathode had a smaller resistance compared to that of the B-Fe₃O₄-C/S cathode.

5. Page 22, in Fig. 5b, the model of SP-Fe₃O₄-C/S and B-Fe₃O₄-C/S looks like the same, except for the distribution of the colors. More detailed descriptions and annotations are needed to clarify the relationship between the models and the two samples employed in this study.

Response: We appreciate this constructive comment. In our study, a model with continuous framework and channel was first constructed and then the channels were filled with sulfur to simulate the SP-Fe₃O₄-C/S cathode. In contrast, another model with discontinuous mesopores was also constructed and then filled with sulfur to simulate the B-Fe₃O₄-C/S cathode. The two models appear to be the same in visual because the pores are filled with sulfur. Actually, the simulations of the current density distribution and the electric field during charging and discharging more clearly reflect the effects of different pore structures of the host materials. To provide a more intuitive description of the two models, we have added more detail information for the models of SP-Fe₃O₄-C/S and B-Fe₃O₄-C/S in the revised manuscript (Page 21, Lines 372-375).

Reviewer #3 (Remarks to the Author):

Comments:

This manuscript demonstrated a kind of Fe₃O₄-doped carbon material with single primitive bicontinuous topology as a host of sulfur cathode in Li-S batteries. The authors combined the experiments with finite element simulation to illustrate that this cathode shows good ion transport capability and fast kinetics. And this cathode also displays good cycle life over 1200 at 1 C. However, there are some questions need to be further addressed.

1. In Figure 4d, we can see the linear fit of A2 peak is not good. Maybe the relationship between the peak current of A2 and scan rate do not follow the Randle-Sevcik equation.

Response: We appreciate the reviewer's carefulness. We give our explanations here based on several evidence. (1) Actually, in experiment, the peaks and scan rates were hard to fit perfectly with the theoretical Randle-Sevcik equation. Since real experiments are affected by many factors (e.g. temperature changes, environment around the material, production process, etc.), it is easy to understand that there is a certain amount of error between the experiment and the theory. In our experimental results, we could still find the linear trend after fitting in Fig. R1. Similar situations were also found in many reports (e.g. *Joule* **2018**, 2, 2091; *Chem. Eng. J.* **2022**, 430, 132677). In our study, the slope of peak A2 is obviously lower than that of peak A1, which does not affect the conclusion. (2) Obviously, R² value of the fitting curve of peak A2 is closed to 0.99 in Table R1 (the specific fitting method is provided in the caption of Table R1); it is generally believed that fitting with an R² value near 0.99 is credible (*Nat. Commun.* **2020**, 11, 436). Therefore, we believe that the linear fit of peak A2 is reliable.

Fig. R1 **a** Representative voltammograms of SP-Fe₃O₄-C/S at different scan rates. **b** Representative voltammograms of B-Fe₃O₄-C/S at different scan rates. **d, e** Peak currents versus the square roots of the scan rates of SP-Fe₃O₄-C/S (peak A1, B1, C1) and B-Fe₃O₄-C/S (peak A2, B2, C2), respectively.

Table R1. Linear fit parameters of the peak currents versus the square roots of the scan rates (Line fitting equation: $y = a + bx$; R^2 is the correlation coefficient, reflecting the degree of fit of the regression model and the reliability of the data).

Peaks	A1	A2	B1	B2	C1	C2
a	0.11529 ± 0.00114	0.06258 ± 0.0095	$0.07679 \pm 8.865 \times 10^{-4}$	0.06087 ± 0.00171	$0.18554 \pm 1.51864 \times 10^{-17}$	0.17419 ± 0.00394
b	$2.42524 \times 10^{-4} \pm 1.97416 \times 10^{-5}$	$8.29252 \times 10^{-6} \pm 1.64511 \times 10^{-4}$	$-3.067 \times 10^{-4} \pm 1.535 \times 10^{-5}$	$-3.67 \times 10^{-4} \pm 2.96 \times 10^{-5}$	$-1.43038 \times 10^{-4} \pm 2.51631 \times 10^{-19}$	$-5.97909 \times 10^{-4} \pm 6.82012 \times 10^{-5}$
R^2	0.9997	0.9871	0.9996	0.9976	1.0	0.9985

2. The authors claimed that the channels of SP-Fe₃O₄-C were almost completely filled with sulfur. However, in Figure S10a-c, we cannot know whether the sulfur can be filled in the pores deep inside the structure. And in the BET test, if the pores of the outer surface of the structure are filled with sulfur completely but the inside pores not, the surface area will also be low due to the blocking of the sulfur on the surface. Please provide more data to confirm that the channels of SP-

Fe₃O₄-C were almost completely filled with sulfur.

Response: This concern is excellent. Actually, we have considered this issue. We give our explanations here based on several evidence. (1) The sulfur melt-diffusion method, which we applied in our study, was a general method for loading sulfur into carbon-based porous materials (*Nat. Mater.* **2009**, *8*, 500; *Adv. Mater.* **2022**, *34*, 2108363; *Nat. Nanotechnol.* **2021**, *16*, 166). The sulfur loading process in our work was carried out at 155 °C for 12 h. At this temperature, the sulfur was liquid and could penetrate deeply into the 3D continuous channels in SP-Fe₃O₄-C. (2) As a line of evidence, the mesochannels cannot be observed under TEM after the sulfur loading (see the comparison in Fig. R2 or the revised Supplementary Fig. 10b,d). If the sulfur only covered on the surface, the mesopores in the TEM image can still be identified. Moreover, as the mapping image in the revised Supplementary Fig. 10e shows, the element sulfur distributes homogeneously in the particle. (3) Based on the TGA result, an ultra-high sulfur loading content of 75 wt% was achieved (Supplementary Fig. 10f). (4) According to the results of nitrogen adsorption-desorption analysis, the specific surface areas (SSAs) of SP-Fe₃O₄-C sharply decreased from 485 to 49 m²g⁻¹, and pore volume also had a large decrease (1.08 to 0.31 cm³ g⁻¹) (Supplementary Fig. 10h,i; Supplementary Table. 1). Although the BET and TGA data provide indirect evidence for the sulfur loading, 75wt% sulfur cannot distribute merely on the SP-Fe₃O₄-C surface. (5) For comparison, the cycling performances under low/high rates and high surface sulfur loading were put in Fig. R3. All of the excellent cycling performances of the SP-Fe₃O₄-C/S-based Li-S batteries reflected that most of sulfur was loaded inside the mesochannels of SP-Fe₃O₄-C. If the sulfur only covered on the surface, the shuttle effect could not be avoided and the battery could not achieve such a remarkable electrochemical performance (especially the long cycle life). Based on these results, we believe that the sulfur have been filled in the whole SP-Fe₃O₄-C particles. In order to clarify our viewpoint more clearly, we have also revised Supplementary Fig. 10 in the revised SI.

Fig. R2 **a** TEM image of a SP-Fe₃O₄-C particle (the inset is a model of SP-Fe₃O₄-C). **b** TEM image of a SP-Fe₃O₄-C/S particle (the inset is a model of SP-Fe₃O₄-C/S). **c** Magnified view of SP-Fe₃O₄-C (clear open channels are observed in SP-Fe₃O₄-C). **d** Magnified view of SP-Fe₃O₄-C/S (open pores are invisible).

Fig. R3 Electrochemical comparison of the SP-Fe₃O₄-C/S-based and B-Fe₃O₄-C/S-based Li-S batteries. **a** Cycling performance at 0.2 C. **b** Long cycling performance at 1 C. **c** Cycling performance of the SP-Fe₃O₄-C/S-based Li-S batteries with sulfur loading of 3.2, 5.6 and 8.2 mg cm⁻², respectively.

3. In page 16, the authors mentioned that the high conductivity of Fe₃O₄ facilitated charge transfer. What does high conductivity mean? Fe₃O₄ is only a kind of semiconductor.

Response: The reviewer's concern is reasonable. The electrical conductivity of Fe₃O₄ is about $5 \times 10^4 \text{ S m}^{-1}$ (*Appl. Phys. Lett.*, **1998**, 72, 734; *Adv. Mater.* **2017**, 29, 1702707), which is higher than those of most common metal oxides and is sufficient for charge transfer during battery operation (*Adv. Energy Mater.* **2018**, 8, 1800595; *Angew. Chem., Int. Ed.* **2015**, 54, 12886; *Adv. Funct. Mater.* **2017**, 27, 1604265; *Science* **2021**, 373, 1494). We have revised the the description “high conductivity“ to “certain conductivity“ in the revised manuscript (Page 4, Lines 77, 78).

4. There are some mistakes in this manuscript. In Page 23, Line 402, Li₂S should be corrected to Li₂S. The authors should check the whole manuscript carefully.

Response: We appreciate the reviewer's carefulness. We have revised it in the revised manuscript and checked the whole manuscript carefully.

5. In Figure 4(i3), during discharge, we can see a new semicircle formation in the EIS plot. Please explain the reason.

Response: This is a very helpful reminding. To our knowledge, two depressed semicircles in high frequency and middle-high frequency regions can be attributed to the interface resistance and charge transfer resistance, respectively (*Adv. Energy Mater.* **2022**, *12*, 2102774; *Nat. Commun.* **2023**, *14*, 4474; *J. Electrochem. Soc.* **2019**, *166*, A5090). We considered that the formation of the new semicircle in the middle-high frequency region of EIS plot could be related to the charge transfer resistance. Before discharge, the introduction of the bicontinuous structure decreased both the interface resistance and charge transfer resistance, reflecting the advantage of the structure.

6. Whether Fe₃O₄ shows some redox reactions in the operating voltage range of Li-S battery.

Response: This is also an excellent concern. We give our explanations from the following two aspects: (1) The operating voltage of Li-S battery is generally 1.7~2.8 V (*Nat. Nanotechnol.* **2021**, *16*, 166). In most of the reported Fe₃O₄-based lithium batteries, two well-defined reduction peaks could be found at 0.67 and 0.27 V in the first discharge process of cyclic voltammetry (CV), which corresponded to the two electroreduction processes (reactions 1 and 2, e.g. *Nano Lett.* **2013**, *13*, 6136; *ACS Nano* **2013**, *7*, 4459). In the following equations, Fe₃O₄ are written as (Fe³⁺)[Fe₂^{3+/2+}]O₄.

Both of the two reduction peaks (0.67 and 0.27 V) were much lower than 1.7 V (*Nano Lett.* **2013**, *13*, 6136; *ACS Nano* **2013**, *7*, 4459; *Adv. Energy Mater.* **2016**, *6*, 1600256). Since Fe₃O₄ undergoes redox reaction at the voltage of 0.67 or 0.27 V, Fe₃O₄ will not show redox reaction at the operating voltage window of Li-S battery (1.7~2.8 V). (2) In most of the reported Fe₃O₄-based Li-S batteries in literature (e.g. *Chem. Eng. J.* **2021**, *410*, 128153; *J. Mater. Chem. A* **2019**, *7*, 21747), no redox peaks appeared in the CV curves of cathodes without active sulfur species, indicating that the batteries cannot provide capacity without the presence of active sulfur species;

in another word, the sulfur species, rather than Fe_3O_4 , are the active materials. Therefore, we consider that in our study, Fe_3O_4 does not any show redox reactions in the operating voltage range of Li-S battery and it only serves as the adsorption sites and a catalyst.

REVIEWERS' COMMENTS

Reviewer #1 (Remarks to the Author):

All comments addressed and the manuscript can be published as it is now.

Reviewer #2 (Remarks to the Author):

Previous questions have been addressed, it is now recommended for acceptance.

Reviewer #3 (Remarks to the Author):

The author answered all of my questions and I think it can be published in this journal.